



# An investigation into the chemistry of HONO in the marine boundary layer at Tudor Hill Marine Atmospheric Observatory in Bermuda

Yuting Zhu[1], Youfeng Wang[1,2], Xianliang Zhou[1,3], Yasin Elshorbany[4], Chunxiang Ye[2], Matthew Hayden[5], Andrew J. Peters[5]

1. Wadsworth Center, New York State Department of Health, Albany, NY, United States

2. State Key Laboratory of Environmental Simulation and Pollution Control, College of Environmental Sciences and Engineering, Peking University, Beijing, China

3. Department of Environmental Health Sciences, University at Albany, State University of New York, NY, United States

4. Atmospheric Chemistry and Climate Laboratory, College of Arts & Sciences, University of South Florida, St. Petersburg, FL, United States

5. Bermuda Institute of Ocean Sciences, St George's, Bermuda

*Correspondence to:* Yuting Zhu (yzhu@syr.edu) and Xianliang Zhou (xianliang.zhou@health.ny.gov)



**Abstract**

Here we present measurement results of temporal distributions of nitrous acid (HONO) along with several
chemical and meteorological parameters during the spring and the late summer of 2019 at Tudor Hill Marine
Atmospheric Observatory in Bermuda. Large temporal variations in HONO concentration were controlled by
several factors including local pollutant emissions, air mass interaction with the island, and long-range
atmospheric transport of HONO precursors. In polluted plumes emitted from local traffic, power plant and

cruise ship emissions, HONO and nitrogen oxides ($NO_x$) existed at substantial levels (up to 278 pptv and 48
ppbv, respectively) and $NO_x$-related reactions played dominant roles in daytime formation of HONO. The
lowest concentration of HONO was observed in marine air, with median concentrations at ~3 pptv around solar
noon and <1 pptv during the nighttime. Considerably higher levels of HONO were observed during the day in
the low-$NO_x$ island-influenced air ($[NO_2]$ < 1 ppbv), with a median HONO concentration of ~17 pptv. HONO

mixing ratios exhibited distinct diurnal cycles that peaked around solar noon and were lowest before sunrise,
indicating the importance of photochemical processes for HONO formation. In clean marine air, $NO_x$-related
reactions contributed to ~35% of the daytime HONO source and the photolysis of particulate nitrate ($pNO_3$) can
account for the missing source assuming a moderate enhancement factor of 30 relative to gaseous nitric acid
photolysis. In low-$NO_x$ island-influenced air, the contribution from both $NO_x$-related reactions and $pNO_3$

photolysis accounted for only ~30% of the daytime HONO production, and the photochemical processes on
surfaces of the island, such as the photolysis of nitric acid on the forest canopy, might contributed significantly
to the daytime HONO production. The concentrations of HONO, $NO_x$ and $pNO_3$ were lower when the site was
dominated by the aged marine air in the summer and were higher when the site was dominated by North
American air in the spring, reflecting the effects of long-range transport on the reactive nitrogen chemistry in

the background marine environments.



## 1 Introduction

Nitrous acid (HONO) is a reactive nitrogen species that plays an important role in the oxidation capacity of the troposphere, as its rapid photolysis (R1) can account for a significant fraction of the production of hydroxyl

radical (OH) (Elshorbany et al., 2010, 2012; Kleffmann et al., 2005; Perner and Platt, 1979):

$$HONO \xrightarrow{h\nu} NO + OH \qquad \text{(R1)}$$

The chemistry of HONO, especially during the daytime, is not well understood. Known HONO sources include direct emissions (Li et al., 2008b; Oswald et al., 2013; Su et al., 2011; Trentmann et al., 2003), reactions between nitrogen oxide (NO) and OH (Zabarnick, 1993), heterogeneous production from nitrogen dioxide ($NO_2$)

(Kleffmann, 2007), and the photolysis of nitrate on surfaces (Zhou et al., 2003) or in the aerosol phase (Ye et al., 2017b):

$$NO + OH \xrightarrow{M} HONO \qquad \text{(R2)}$$

$$NO_2 + H_2O, \text{organics} \xrightarrow{\text{surface}} HONO \qquad \text{(R3)}$$

$$HNO_3(s) + H_2O, \text{organics} \xrightarrow{h\nu} HONO + NO_x \qquad \text{(R4)}$$


$$pNO_3(s) \xrightarrow{h\nu} HONO + NO_x \qquad \text{(R5)}$$

Recent field observations reported that HONO exists at considerable levels during the day, up to several parts-per-billion volume (ppbv) in the urban atmosphere (e.g., Acker et al., 2006; Elshorbany et al., 2009) and up to several hundred parts-per-trillion volume (pptv) in the rural and remote atmosphere (He et al., 2006; Ye et al., 2016; Zhang et al., 2009). During the daytime, HONO is removed rapidly by its photolysis, with short

photolytic lifetime of ~10 min at noon under clear-sky conditions (Elshorbany et al., 2009; Kleffmann et al., 2003; Zhang et al., 2009). To sustain the observed HONO concentration against the major photochemical loss of HONO, active daytime productions are needed to explain its observed mixing ratios (Zhou et al., 2001, 2002).

In the urban environment, direct HONO emission only accounts for a small fraction of its observed atmospheric concentration (Kleffmann, 2007). It is commonly believed that HONO is mainly formed by the gas-phase

reactions between NO and OH (Elshorbany et al., 2010; Villena et al., 2011b) and heterogeneous reactions of $NO_2$ (Finlayson-Pitts et al., 2003; Kleffmann, 2007). The heterogeneous reactions leading to HONO formation occur on a variety of surfaces, including aerosol, soot, vegetation, and urban surfaces (He et al., 2006; Kleffmann et al., 1999; Ramazan et al., 2004; Reisinger, 2000; Stutz et al., 2002). It was also suggested that HONO formations are likely enhanced by solar irradiation, since measurement studies observed unexpectedly

high HONO concentrations during the daytime in various environments (Acker et al., 2006b; Kleffmann et al., 2005; Vogel et al., 2003). One possible explanation was that organic matter serve as photosensitizers and result in light-initiated enhancement of HONO formation during the hydrolysis of $NO_2$ (George et al., 2005; Stemmler et al., 2006, 2007). Other proposed mechanisms for HONO production include gas-phase reactions of excited $NO_2$ with water (Li et al., 2008a) and the reactions between $NO_2$ and the hydroperoxyl-water complex



(HO$_2$·H$_2$O) (Li et al., 2014); recent model calculations suggested that these newly proposed, NO$_2$-related mechanisms are likely insignificant as HONO precursors (Carr et al., 2009; Wong et al., 2011; Ye et al., 2015).

In atmospheric environments with relatively low abundance of NO$_x$ (NO$_x$ = NO + NO$_2$), studies proposed that rapid photolysis of HNO$_3$ on various surfaces as well as particulate nitrate (pNO$_3$) photolysis might serve as important HONO sources (Ye et al., 2016; Zhou et al., 2003, 2011). Direct evidence from laboratory

photochemistry experiment showed that there exist major photolytic rate enhancements of HNO$_3$ on various surfaces (Ye et al., 2017; Zhou et al., 2003) and in aerosol sample collected from various air masses, including boundary layer air above the Atlantic Ocean and over the terrestrial environments over the United States (Ye et al., 2017b, 2018). Reported pNO$_3$ photolytic rate constant ($J^N_{pNO_3}$, normalized by solar irradiation at tropical noontime) was higher than the rate constant for gaseous nitric acid photolysis by ~1–3 orders of magnitude (Ye

et al., 2017b, 2018).

To date, few studies have investigated HONO distribution and its chemical cycling in the marine boundary layer (MBL), even though the oceans cover >70% of the earth surface (Kasibhatla et al., 2018; Reed et al., 2017; Wojtal et al., 2011; Ye et al., 2016; Zha et al., 2014). Several studies observed substantial HONO concentrations (~ 3.5–11 pptv) during the day in clean, well-aged marine air (Kasibhatla et al., 2018; Reed et al., 2017; Ye et

al., 2016). Evidences from both field observations and laboratory experiments suggested that pNO$_3$ photolysis might contribute to a major fraction of the observed level of HONO in marine atmosphere (Ye et al., 2016). However, major uncertainties persist in current model evaluations for the impact of pNO$_3$ photochemistry on reactive nitrogen cycling in the MBL, due to the paucity of field measurements as well as laboratory experiments that determined $J^N_{pNO_3}$ in marine aerosol samples. Additionally, the role of the ocean surface in

regulating HONO cycling in the MBL remains unclear. Ocean surface is expected as a HONO sink due to its high solubility in alkaline aqueous solutions. However, several studies reported nighttime accumulation of HONO in the MBL, with higher NO$_2$ to HONO conversion rate in air masses that have passed over the ocean surface than those that have passed over the terrestrial surface (Wojtal et al., 2011; Zha et al., 2014). It is possible that there exists an unknown HONO source at the air-sea interface; further evidence in the field and the

laboratory is required to confirm this potential HONO source.

During the spring and late summer of 2019, we conducted extensive field measurements to determine the temporal distributions of HONO and the relevant parameters including NO$_x$, pNO$_3$, HNO$_3$, aerosol loading, O$_3$, and meteorological parameters, at Tudor Hill Marine Atmospheric Observatory (THMAO) in Bermuda. To examine the importance of pNO$_3$ as photolytic HONO source, we collected aerosol samples during the field

campaigns and conducted photochemistry experiments to determine the pNO$_3$ photolysis rate constants leading to HONO formation in the gas phase. In the present study, we present results from both field observations and laboratory experiments and discuss the chemistry of HONO as well as other reactive nitrogen species in the marine atmosphere in Bermuda.



## 2 Experimental

### 2.1 Field Observations

Two intensive field studies were conducted at the THMAO site in Bermuda during the spring (April 17 – May 13) and late summer (August 16 – September 10) of 2019. The site is located on the western coast of Bermuda (32° 19' N, 64° 45' W) (Fig. S1). The sampling site is equipped with three trailers for in-house measurements and an aluminum sampling tower that is 23 m above the ground and 53 m above the sea level.

Ambient HONO concentrations were continuously measured employing a long-path absorption photometry (LPAP) system. HONO was collected in a 10-turn glass coil sampler with purified water (18.2 MΩ cm) that was obtained from a Barnstead Nanopure Diamond™ water purification system (Thermo Scientific). The collected nitrite was derivatized with a reagent solution with 5 mM sulfanilamide (SA), 0.5 mM N-(1-Naphthyl) ethylenediamine (NED) and 40 mM hydrochloric acid (HCl). The formed azo dye was detected at 540 nm using a four-channel optic fiber spectrometer (LEDspec, World Precision Instruments) with a 1-m liquid waveguide capillary cell (World Precision Instruments). To minimize potential artifact for HONO measurements on the sampling inlet wall, the HONO measurement system was installed directly on platform of the sampling tower. The sampling assembly consisted of a coil sampler, a 3-way Teflon solenoid valve, and a $Na_2CO_3$ denuder. Background correction was conducted using the "zero-HONO" air that was generated by pulling ambient air through the $Na_2CO_3$ denuder. The LPAP system ran 60 min alternating cycles; during each cycle the solenoid valve feeds the coil sampler with "zero-HONO" air for 10 min and ambient air for 50 min. The absorbance signal was recorded in every 5 sec; data were averaged in every 10 minutes for further analyses. The LPAP system has successfully proven its effectiveness and accuracy in HONO measurement during the previous NOMADSS field data, and the reader is referred to Ye et al. (2016b, 2018) for detailed discussion for method validation and interference elimination. Briefly, interference species $NO_x$, peroxylacetyl nitrate (PAN) and particulate nitrite are expected to pass through the $Na_2CO_3$ denuder and therefore was subtracted from the ambient HONO signals. Interference from peroxynitric acid ($HO_2NO_2$) is expected to be negligible due to the low steady-state concentration at warm temperature (i.e., > 20 °C). Air sampling rate was 3 L min⁻¹. Detection limit of HONO was calculated as three times of standard deviation (3σ) for baseline signal and was ~0.6 pptv at 10-min time resolution.

Ambient $HNO_3$ and $pNO_3$ was measured by LPAP systems similar to that for HONO and were installed directly on the platform of the sampling tower. Gaseous $HNO_3$ and HONO was collected in a 10-turn glass coil sampler with purified water, the collected nitrate was converted into nitrite by passing through a Cd column in $NH_4Cl$ buffer solution. "zero-HONO/ $HNO_3$" air was generated by pulling ambient air through a $Na_2CO_3$ denuder. HONO concentration measured by the first LPAP system was used to calculate the absorption signal contributed by HONO, which was subtracted from the total absorption signal to obtain the final product for $HNO_3$ concentration. Air sampling rate was 2 L min⁻¹. Detection limit of $HNO_3$ (3σ) was ~2 pptv at 10-min time resolution. Aerosol nitrate was scrubbed from ambient air in a continuously wetted frit-disc sampler with purified water. Collected nitrate was converted into nitrite by passing through a Cd column in $NH_4Cl$ buffer



solution. A $Na_2CO_3$ denuder was installed right before the frit-disc sampler to remove HONO and $HNO_3$. "zero-$pNO_3$" air was generated by pumping ambient air through a 0.45-µm Teflon filter (Sartorius Biolab Products, 47 mm diameter). Air sampling rate was 2 L $min^{-1}$. Detection limit of $pNO_3$ ($3\sigma$) was ~17 pptv at 10-min time resolution. A correction factor of 2.06 was applied to $pNO_3$ concentrations that were measured by the LPAP

system, after comparing the data with the aerosol nitrate concentrations determined in bulk aerosol samples (see Sect. 2.2 for details). The discrepancy likely resulted from loss of aerosol particles within the sampling compartment of the LPAP system (e.g., deposition on the walls of the $Na_2CO_3$ denuder) before the aerosol samples were scrubbed in the wetted frit-disc sampler. After the correction factor was applied, good agreements were achieved when comparing the temporal trends of particulate nitrate concentrations determined by the two

different methods (Fig. 1).

$NO_x$ concentrations were measured by low-level commercial chemiluminescence analyzer (Thermo Environmental Instruments Inc. 42C-TL) with a blue-light $NO_2$ to NO convertor (Droplet Measurement Technologies); detection limit ($3\sigma$) was ~88 and 30 pptv for $NO_2$ and NO, respectively at 10-min resolution. Measurement data from the $NO_x$ analyzer meet our needs in general examination of temporal data variations that

were largely influenced by pollution episodes (see Sect. 3.1 and 3.2). However, $NO_x$ levels in relatively clean marine air were near or lower than the detection limits of the $NO_x$ analyzer. Therefore, data obtained by the $NO_x$ analyzer were not applicable in advanced data analyses (see Sect. 3.3 and 3.4), especially in clean marine air. During the summer field campaign, we installed a $NO_2$ measurement system employing LPAP technique ($NO_2$-LPAP). The $NO_2$-LPAP system was assembled following Villena et al., (2011a). In our customized $NO_2$-LPAP

system, ozone ($O_3$) was firstly removed with 0.6 g/L potassium indigotrisulfonate (Villena et al., 2011a) in a 10-turn coil sampler, the $O_3$-free air later passed a 50-turn coil sampler that scrubbed $NO_2$ with a 14g/L SA (dissolved in 2.5 M acetic acid) solution and a 0.5 g/L NED solution. Air sampling rate was 0.5 L $min^{-1}$. $NO_2$-LPAP measurements (only available in the summer field campaign) successfully lowered the detection limit ($3\sigma$) to 14 pptv at 10-min resolution. Measurement systems for $NO_x$, $NO_2$-LPAP, and $O_3$ were installed in the

trailers; ~30-m long Teflon tubing was used to sample ambient air from the platform of the sampling tower to the in-house measurement systems. $O_3$ was monitored with a commercial ozone monitor (Thermo 49i) based on UV absorbance at 254 nm, and the detection limit ($3\sigma$) was 1.7 ppbv at 10-min resolution.

Aerosol loading was monitored by a commercial nephelometer (Thermo Scientific) which was installed just below the platform of the sampling tower; the nephelometer provide a detection resolution of 0.1 µg $m^{-3}$ and a

particle size range of 0.1–10 µm. The intensity of ultra-violet (UV) light was measured by an Eppley TUV radiometer (295–385 nm) that was installed on the platform of the sampling tower, the measured UV data was later combined with the tropospheric ultraviolet and visible radiative transfer model (https://www2.acom.ucar.edu/modeling/tropospheric-ultraviolet-and-visible-tuv-radiation-model) to calculate the photolysis rate constant for $NO_2$, HONO, $HNO_3$ and $O_3$. Photolysis rate constant calculations were

performed following Zhou et al. (2011), and the details for these calculations are presented in the Supplemental Information.



### 2.2 pNO₃ Photolysis Rate Constant Determination

Bulk aerosol samples were collected on Teflon filters (Sartorius Biolab Products, 0.45 μm pore size, 47 mm diameter) during the field campaigns. There were four sampling period during each day, from 4:00–8:00, 8:00–

12:00, 12:00–16:00 and 16:00–20:00 at local time. Aerosol sampling rate was 10 L min$^{-1}$. The collected aerosol samples were stored in tightly capped petri dishes (Analyslide®, Pall Corporation) in a freezer until use for laboratory experiments. Upon usage, each filter was cut into to two equal halves. One half of the aerosol filter was extracted by 5 mL purified water to determine the concentrations of particulate nitrate (NITs) and other ionic components (Na$^+$, Mg$^{2+}$, Ca$^{2+}$, NH$_4^+$, Cl$^-$, SO$_4^{2-}$) using a Dionex ICS 3000 Ion Chromatography System

coupled with an AS14 4 mm analytical common and an AG14 4 mm guard column.

The other half filter was used in light exposure experiment to determine the production rate constant of HONO. During each photochemistry experiment, the aerosol filter was placed in a cylindrical flow reactor that was installed under a solar simulator. The cylindrical flow reactor, with 10-cm diameter and 1.5-cm depth, was made from a Teflon block with a quartz window on the top. A 300 W Cermax® Xenon lamp (Perkin Elmer) served as

the light source in the solar simulator. Humidified zero air (50% RH) was used as the carrier gas that flow through the Teflon block reactor and the flow rate was 1 L min$^{-1}$. The amount of HONO that was produced during short-period (5 min) light exposure was determined with a LPAP system that was installed after the flow reactor. The readers are referred to the Supplemental Information for detailed calculations of aerosol nitrate photolysis rate constant leading to HONO production $J_{pNO_3}^N$, which was normalized to light conditions during

tropical summer with solar zenith angle of 0°.

### 3 Results and Discussion

Fig. 1 presents an overview for some chemical and meteorological parameters that were measured during the spring and late summer field campaigns of 2019, including HONO, NO$_x$, pNO₃, NITs, J$_{HONO}$, wind speed and wind direction. Data statistics for HONO, NO$_x$, pNO₃, NITs concentrations and HONO/NO$_x$ ratios were

summarized in Table 1. As shown in Fig. 1, we observed large temporal variations for the concentrations of reactive nitrogen species. These temporal variations resulted from joint influences by local pollutant emissions (see Sect. 3.1 for detailed discussion), long-range transport of air masses (see Sect. 3.2 for detailed discussion) and diurnal changes in solar radiation (see Sect. 3.3 for detailed discussion).






**Table 1:** Data statistics for HONO, NO$_x$, pNO$_3$, NITs and HONO/NO$_x$ ratio data during the spring (April 17 – May 13) and late summer (August 16 – September 10) of 2019 at the THMAO in Bermuda. For HONO, NO$_x$, pNO$_3$ and HONO/NO$_x$ ratio, the statistical analyses were performed based on 10-min averaged data.

| | | HONO (pptv) | NO$_x$ (pptv) | pNO$_3$ (pptv) | NITs (pptv) | HONO/NO$_x$ |
|---|---|---|---|---|---|---|
| Spring | Range[a] | 0.5 – 34.1 | bld[b] – 2084 | 41.1 – 789 | 77.9 – 811 | bld[b] – 0.13 |
| | Mean ± SD | 9.8 ± 12 | 606 ± 1970 | 311 ± 227 | 300 ± 235 | 0.030 ± 0.24 |
| | Median | 5.4 | 186 | 284 | 219 | 0.023 |
| | N | 2600 | 3389 | 2116 | 93 | 2300 |
| Summer | Range[a] | bld[b] – 19.9 | bld[b] –560 | 32.6 – 235 | 50.1 – 302 | bld[b] – 0.21 |
| | Mean ± SD | 6.7 ± 12 | 228 ± 866 | 130 ± 145 | 157 ± 101 | 0.048 ± 1.3 |
| | Median | 3.6 | 90.3 | 120 | 126 | 0.025 |
| | N | 3221 | 3283 | 2809 | 93 | 2889 |

[a] 5th to 95th percentiles were used to represent the data ranges.

[b] bld denotes that the concentrations were below the limit of detection.





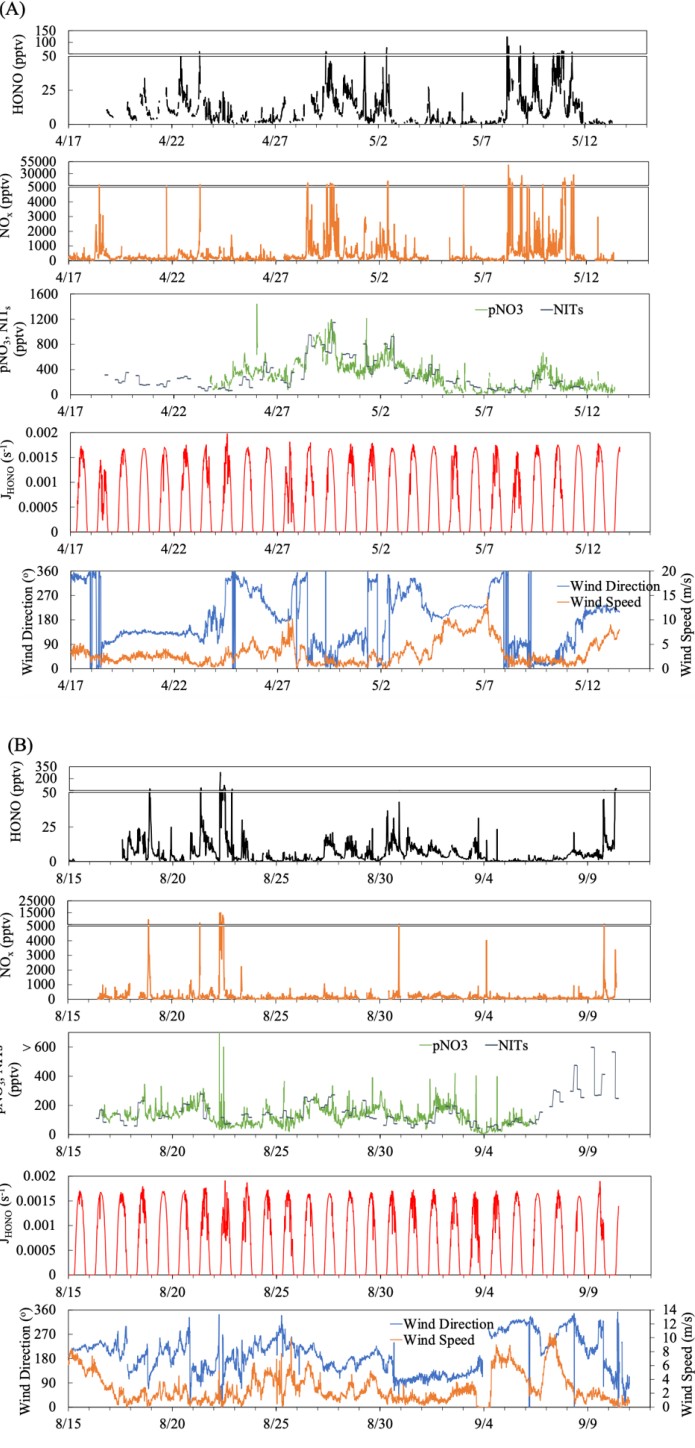



**Figure 1:** Time series plots of HONO, NO$_x$, pNO$_3$, NITs, HONO photolysis rate constant (J$_{HONO}$) wind direction
and wind speed for the spring (A) and summer (B) field campaigns that were conducted at the THMAO in
Bermuda.

### 3.1 Impact from local emissions

At the THMAO, reactive nitrogen measurements were quite sensitive to directions of local winds. Fig. S2 shows
wind rose plots representing the distribution of wind direction and wind speed during the spring and summer
field campaigns. Fig. 2 represents bivariate polar plots (generated with *openair* – an R package for air quality
data analysis, Carslaw and Ropkins, 2012) showing the joint wind speed and wind direction dependences of
HONO, NO$_x$, pNO$_3$ and HONO/NO$_x$ ratios. When prevailing winds blew from the northwest or southwest, clean
marine air with relatively low concentrations of HONO and NO$_x$ were brought to the sampling site. It was also
shown by Fig. 2 that urban and power plant plumes transported to the THMAO site with northeasterly winds.
These plumes, originated from the city of Hamilton (i.e., capital city of Bermuda with busy road traffics and a
nearby power plant) and the Royal Naval Dockyard (i.e., a harbor of cruise ships), contained HONO and NO$_x$ at
substantial levels (Fig. 2). The highest concentrations of HONO and NO$_x$ reached 278 pptv and 48 ppbv,
respectively. The pollution level within the plumes was comparable with that observed in polluted marine
atmospheric environments at coastal sites in Hong Kong (Zha et al., 2014), on Tuoji Island in eastern Bohai Sea,
China (Wen et al., 2019), and on Saturna Island in British Columbia, Canada (Wojtal et al., 2011).





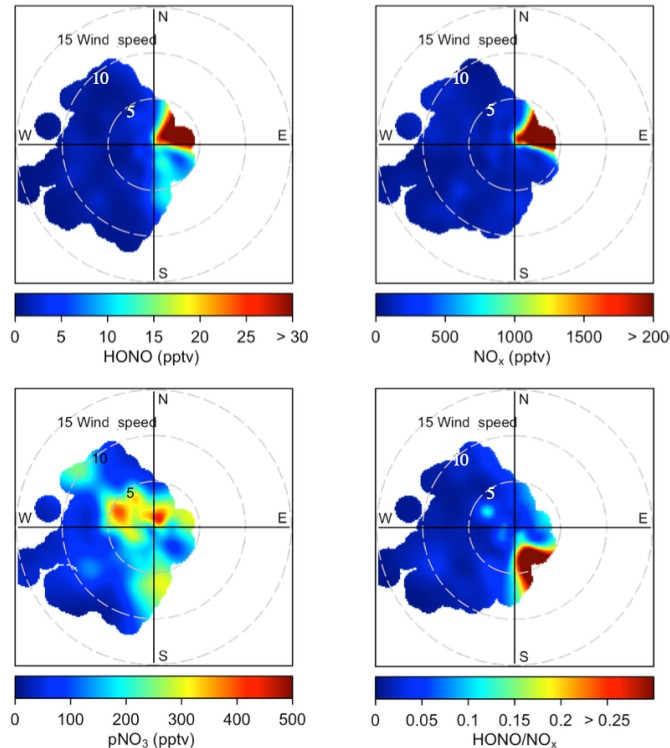

**Figure 2:** Bivariate polar plots showing the joint wind speed and wind direction dependences of HONO, $NO_x$, $pNO_3$ and HONO/$NO_x$ ratios during the two field campaigns in Bermuda. Data measured during the spring and the summer seasons were combined to generate these polar plots. The color scales represent HONO, $NO_x$, $pNO_3$ concentrations, and HONO/$NO_x$ ratios. The radical scale shows the wind speed in m/s.

Here we divided our measurement datasets into different categories based on three types of local influences: clean marine air (wind directions ranged from 180 to 360°), island-influenced air (wind directions ranged from 0 to 180°, [$NO_2$] < 1 ppbv) and polluted plumes (wind directions ranged from 0 to 180°, [$NO_2$] ≥ 1 ppbv). It should be pointed out that the clean marine air was still affected by ship emissions, and thus spikes were removed from the data set when averaging HONO and $NO_2$ concentration in clean marine air to eliminate influences of emissions from passing ships. Whisker plots comparing HONO, $NO_x$, $pNO_3$ concentrations and HONO/$NO_x$ ratios under different types of local influences were presented in Fig. 3. HONO and $NO_x$ concentrations in polluted plumes were higher than those in marine by ~1–3 orders of magnitude. The contribution from direct emissions to the high-level HONO may be substantial during the night but is expected to be relatively small during the day. Estimated transport times from the city of Hamilton to the THMAO site were ≥ 1.4 h (with ~10 km distance and ~2 m/s wind speed) and were several times longer than the photolytic





lifetime of HONO in the daytime. Therefore, the observed HONO in the daytime plumes at the site was mostly

produced during the air transport from emission sources to the THMAO site via gas-phase reaction (R2),

heterogeneous reactions (R3) on aerosol and island surfaces, and maybe some other chemical sources.

The island-influenced air was the marine air mass that passed over less populated regions of the island and was

thus not significantly impacted by anthropogenic emissions. The concentrations of HONO and $NO_x$ were higher

in island-influenced air than in clean marine air (Fig. 3). It was also observed that the highest HONO/$NO_x$ ratios

(an indicator for the efficiency of HONO production) were found in island-influenced air. The underlying

mechanism for this active HONO production is unknown. One possible explanation is that reactions occurring

on various surfaces (e.g., the ground and vegetation surfaces) of the island contribute significantly to HONO

production during the day (see Sect. 3.4 for detailed discussion).

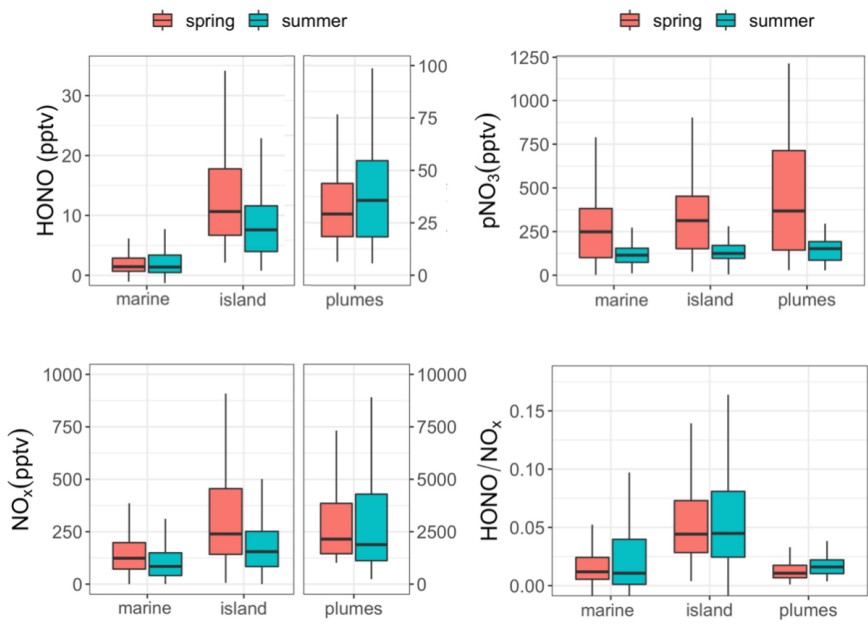


**Figure 3:** Whisker plots comparing of HONO, $NO_x$, $pNO_3$ concentrations and HONO/$NO_x$ ratios in clean marine

air, island-influenced air, and polluted plumes during the spring and the summer field campaigns in Bermuda.

The thick lines represent median values, the boxes the 25th and 75th percentiles, and the whiskers the largest

value within 1.5 times interquartile range above 75th percentile and the smallest value within 1.5 times

interquartile range below 25th percentile.

Fig. 3 also showed that local wind direction did not appear as a significant factor controlling $pNO_3$

concentrations. It was concluded that $pNO_3$ concentration was mainly controlled by the origins of the air masses





and long-range transport (see Sect. 3.2) and was relatively independent from the impacts by local pollution
emissions.

### 3.2 Impact from long-range transport

The THMAO, located on the western coast of the Bermuda island, is influenced by air masses that originated
from different source regions (Moody and Galloway, 1988; Todd et al., 2003). Fig. S3 showed a summary of the
7-day backward trajectories; the trajectories were calculated daily starting at local noontime using NOAA'S

HYSPLIT model (https://www.ready.noaa.gov). In the spring season, 23 out of 27 of the trajectories indicated
northwesterly flows that originated from the North American continent. Exceptions were found from May 3 to
May 7 when the sampling site received aged marine air. In the summer season, the site was under the influence
of the Bermuda high and received well-aged marine air above the Atlantic Ocean. On September 7, the Bermuda
high was disturbed as hurricane Dorian traveled along the eastern coast of the US. Calculated backward

trajectories indicated fast-traveling air flows originated on the North American continent during September 7–
10.

In order to provide direct comparisons of data based on source regions of air masses, i.e., North American (NA)
in the spring and North Atlantic Ocean (NAO) in the summer, we excluded measurement data collected during
May 3–7 and September 7–10, and generated whisker plots (Fig. S4) comparing HONO, $NO_x$, $pNO_3$

concentrations and HONO/$NO_x$ ratios. The plots in Figs. 3 and S4 are almost identical, due to the fact that most of
the air masses originated from North America in spring but circulated above the North Atlantic Ocean during the
summer. It appears that the local emission and the island modification of the air masses controlled the levels of
HONO, $NO_x$ and the HONO/$NO_x$ ratio. However, the level of $pNO_3$ was significantly higher in NA than in NAO as
a result of long-distance transport of this species and was less dependent on how the air masses interacting with the

island. For HONO production efficiencies, no significant differences were found in the HONO/$NO_x$ ratios
between the two air mass categories (Fig. S4) under each type of local influences, as expected from HONO
being a short-lived species controlled by in-situ production and rapid loss (Elshorbany et al., 2009; Kleffmann et
al., 2003; Zhang et al., 2009; Zhou et al., 2001, 2002). However, it should be noted that long-distance traveled
air flows could affect HONO chemistry under certain circumstances. In fact, significant changes in reactive

nitrogen chemistry were observed on 9/7–9/8 while aged North Atlantic air was replaced by a fast-traveling
flow from the Northeastern US. The readers are referred to Sect. 3.5 for detailed discussions.

### 3.3 Diurnal variations

Diurnal cycles for HONO, $NO_2$, $pNO_3$ concentrations and HONO/$NO_2$ ratios in marine and island-influenced air
were presented in Fig. 4 after binning measurement data into 1 h intervals. Urban plumes represented high-$NO_x$

air that were emitted from point sources (e.g., power plant emission) or mobile sources (e.g., road traffic
emissions). Diurnal trends in polluted plumes were not shown in Fig. 4 because (1) temporal variability of
reactive nitrogen species in polluted plumes was largely dependent on the source of the pollutant emission and
(2) data collected in polluted plumes were insufficient to represent diurnal cycles for a 24-h time period.



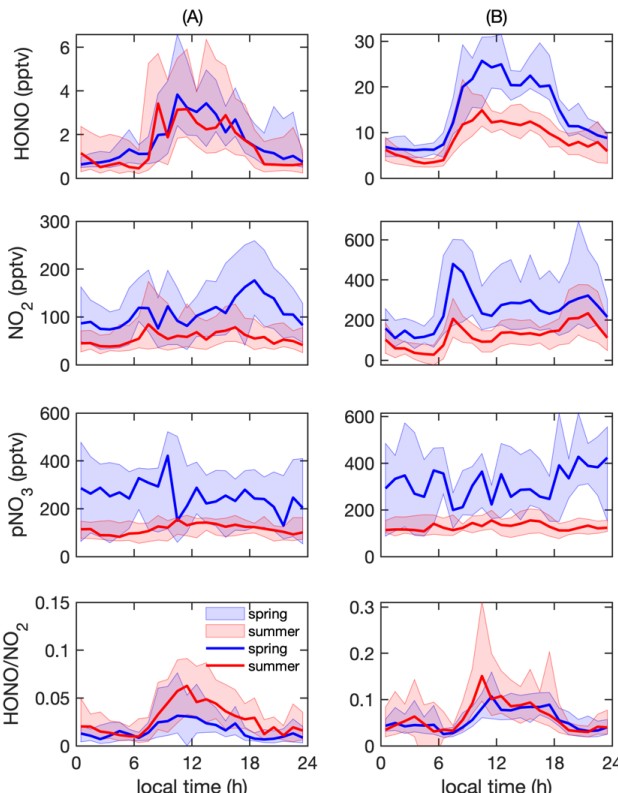

**Figure 4:** Diurnal profiles of HONO (pptv), $NO_2$ (pptv), $pNO_3$ (pptv), and HONO/$NO_2$ ratio in (A) clean marine air and (B) island-influenced air. The solid lines represent the median values and the areas represent $25^{th}$ to $75^{th}$ percentiles.

HONO concentrations exhibited distinct diurnal cycles in clean marine air (Fig. 4A), with maxima of ~3 pptv around solar noon and minima near or below ~1 pptv during the early morning hours before sunrise. The observed diurnal trends of HONO were in good agreement with those observed from Cape Verde Atmospheric Observatory located in the tropical Atlantic boundary layer (Kasibhatla et al., 2018; Reed et al., 2017; Ye et al., 2017a). At night, HONO concentrations reached steady state concentrations of ~1 and 0.7 pptv for the spring and summer season, respectively; this nocturnal HONO concentration steady-states were consistent with previous reports in clean marine boundary layer (Kasibhatla et al., 2018; Reed et al., 2017; Ye et al., 2017a). In island-influenced air, HONO concentrations also exhibited diurnal cycles representing higher concentrations during the daytime than the nighttime. The observed diurnal cycles confirmed that HONO cycling is controlled by local photoproduction and rapid photolytic loss (Elshorbany et al., 2009; Kleffmann et al., 2003; Zhang et al., 2009; Zhou et al., 2001, 2002). During the summer season, median value for noontime maxima of HONO


concentrations was ~13 pptv and was significantly lower than the median daytime maxima of ~23 pptv for the
spring season. This seasonal difference in HONO daytime maxima likely resulted from the low "background"
levels of HONO daytime precursors including $NO_x$ and $pNO_3$ in aged marine air that dominated our summer
field campaign, as discussed in Sect. 3.2.

The concentrations of $NO_2$ were generally higher during the day than during the night, but with a bimodal

distribution pattern that peaks in the morning and the evening hours (Fig. 4), which is different from a
previously reported monomodal pattern of $NO_2$ that peaked around solar noon (Reed et al., 2017; Ye et al.,
2017a). This bimodal distribution pattern, which was more distinct in island-influenced air than in marine air
and also more distinct in spring than in summer, were likely due to the increases in marine traffic and sporting
boat activity in the waters off Bermuda western shore during the westerly air flow; the observed $NO_2$ peaks

coincided with the local morning and evening traffic rush hours during the island-influenced periods, with a
possible 1–2 hour transport time delay from more populated areas. It is interesting to point out that the patterns
of HONO and $NO_2$ were highly different during the day, suggesting that $NO_2$ was unlikely a major HONO
precursor in the background air masses when $NO_x$ was below 1 ppbv.

$pNO_3$ is known as another important HONO precursor (Ye et al., 2016, 2017b, 2018). In both clean marine air

and island-influenced air, we observed significant seasonality and no clear diurnal cycles regarding $pNO_3$
concentrations, confirming the importance of long-range transport contribution to $pNO_3$ concentrations, as
discussed in Sect. 3.2.

The plateaus of HONO concentration and HONO/$NO_2$ ratio consistently occurred in the daytime around solar
noon (±3 h) under all types of local influences, indicating the existence of active daytime photochemical HONO

sources to compensate the rapid photolytic loss of HONO (Oswald et al., 2015; Zhou et al., 2007). The daytime
HONO/$NO_2$ ratios were ~0.04 and 0.10 in clean marine air and island-influenced air, respectively. The
significantly higher HONO/$NO_2$ ratios in island-influenced air is indicative of a significant contribution of
HONO by heterogeneous processes occurring on island surfaces. The readers are referred to Sect. 3.4 for
detailed discussion regarding factors that affected the daytime chemistry of HONO.

During the nighttime, we observed pseudo steady-state concentrations of HONO in the marine atmosphere (Fig.
4). Nighttime steady states in HONO concentrations were previously reported in clean (Kasibhatla et al., 2018;
Reed et al., 2017; Ye et al., 2017a) and polluted marine environment (Wojtal et al., 2011). Several recent
research works showed nighttime HONO accumulations in polluted marine environments (Wen et al., 2019;
Wojtal et al., 2011; Yang et al., 2021; Zha et al., 2014). With higher $NO_2$-to-HONO conversion rates and higher

HONO/$NO_2$ ratios in the air masses passing over sea surface than land surface (Zha et al., 2014), the $NO_2$
heterogeneous reaction on the sea surface microlayer has been proposed to be the nighttime HONO source in the
nocturnal polluted MBL. However, we did not observe any nighttime HONO accumulation in the MBL at
THMAO site in either clean marine air or the island-influenced air. Furthermore, the HONO/$NO_2$ ratios were
lower in the clean marine air (<0.02) than in the island-influenced air (<0.06), contradicting the earlier reports.

The lack of nighttime HONO accumulation and the significantly lower HONO concentrations and HONO/$NO_2$





ratio during the night than the day suggest the existence of a nighttime HONO sink in the absence of its photolytic loss, i.e., the dry deposition of HONO onto ocean surface. It is therefore concluded that ocean surface is a net HONO sink for the clean MBL.

Island-influenced air was a marine air mass that interacted with the land surfaces during its transit over the island before reaching the sampling site. The interaction time between the air mass and the island surface was in the range from less than one hour to several hours, depending on the wind flow direction. Higher pseudo steady-state HONO concentration was quickly reached during the day (Fig. 4) due to its strong photochemical source and fast photolytic loss. During the night, enhanced HONO accumulation from $NO_2$ heterogeneous reactions on aerosol and island surfaces occurred only during the relatively short time when the air mass was over the island. As a result, the nighttime HONO concentration in the island-influenced air would remain low but would be higher than that in the clean marine air (Fig. 4).

### 3.4 Daytime HONO budget

In the current section, we examine daytime HONO budget, equation (1), using the data obtained during the day around solar noon, from 10:00 to 15:00 local time.

$$\frac{d[HONO]}{dt} = P_{HONO} - L_{HONO} \tag{1}$$

where $P_{HONO}$ and $L_{HONO}$ are the overall HONO production and loss rates (in pptv·s⁻¹). During the daytime, known HONO daytime formation sources (Elshorbany et al., 2010; Finlayson-Pitts et al., 2003; Ye et al., 2016; Zhou et al., 2003) include $NO_x$-related reactions (R2, R3) ($P_{NO_x \to HONO}$), the photolysis of $HNO_3$ on surfaces (R4) ($P_{HNO_{3(ads)} \to HONO}$) and the photolysis of pNO₃ (R5) ($P_{pNO_3 \to HONO}$):

$$P_{HONO} = P_{NO_x \to HONO} + P_{pNO_3 \to HONO} + P_{HNO_{3(ads)} \to HONO} + P_{other} \tag{2}$$

where $P_{other}$ represents other unaccounted processes.

HONO photolysis ($L_{photolysis}$) is the dominant HONO sink during the day, accounting for ~95% of HONO loss rate. Other minor HONO sinks include its reaction with OH ($L_{HONO-OH}$) and its dry deposition ($L_{deposition}$):

$$L_{HONO} = L_{photolysis} + L_{HONO-OH} + L_{deposition} \tag{3}$$

In a daytime photo-steady state, $\frac{d[HONO]}{dt} \approx 0$:

$$P_{HONO} \approx L_{HONO} \approx L_{photolysis} = [HONO] \times J_{HONO} \tag{4}$$

Where $[HONO]$ is the concentration of HONO (pptv) and $J_{HONO}$ is the photolysis rate constant of HONO (s⁻¹). Calculated $L_{photolysis}$ were listed in Table 2 for direct comparison with HONO production rates. The median daytime $L_{photolysis}$ values are $4.5 \times 10^{-3}$ pptv·s⁻¹ and $2.4 \times 10^{-2}$ pptv·s⁻¹ in clean marine air and island-influenced air, respectively.



### 3.4.1 NOₓ-related processes

$NO_x$-related processes, including the gaseous reaction between NO and OH (R2) as well as the heterogeneous production from $NO_2$ (R3), were the most well-studied HONO sources during the daytime (Elshorbany et al.,

405    2010; Finlayson-Pitts et al., 2003; He et al., 2006; Kleffmann et al., 1998; Villena et al., 2011b). However, chemical kinetics data on the relevant reactions in the MBL were scarce and thus not sufficient for quantitative analyses of HONO daytime budget. Nighttime HONO accumulation, commonly used to estimate the net $NO_2$ to HONO rates in marine atmospheric environments (Wen et al., 2019; Yang et al., 2021; Zha et al., 2014), was not observed in the present study (see Sect. 3.3). Therefore, we chose to evaluate the role of $NO_x$ as daytime

410    HONO source by extrapolating the $NO_x$-HONO relationship in the high-$NO_x$ plume where $NO_x$ is the dominant precursor via reactions R2–R3. It should be noted that $NO_x$ (NO and $NO_2$) and $NO_2$ were measured simultaneously by the chemiluminescence ($[NO_x]_{chemilum}$) and LPAP techniques ($[NO_2]_{LPAP}$), respectively, during the summer campaign, and that the data quality of $[NO_2]_{LPAP}$ was significantly better than $[NO_x]_{chemilum}$ due to much lower measurement detection limit of the LPAP (14 pptv for 10-min averaged data) than that of

415    chemiluminescence analyzer (88 pptv for 10-min averaged data). Since $[NO_x]$ and $[NO_2]$ were strongly correlated ($R^2 = 0.97$), both sets of data were used to evaluate the HONO production from $NO_x$ reactions (R2–R3) in Fig. 5.



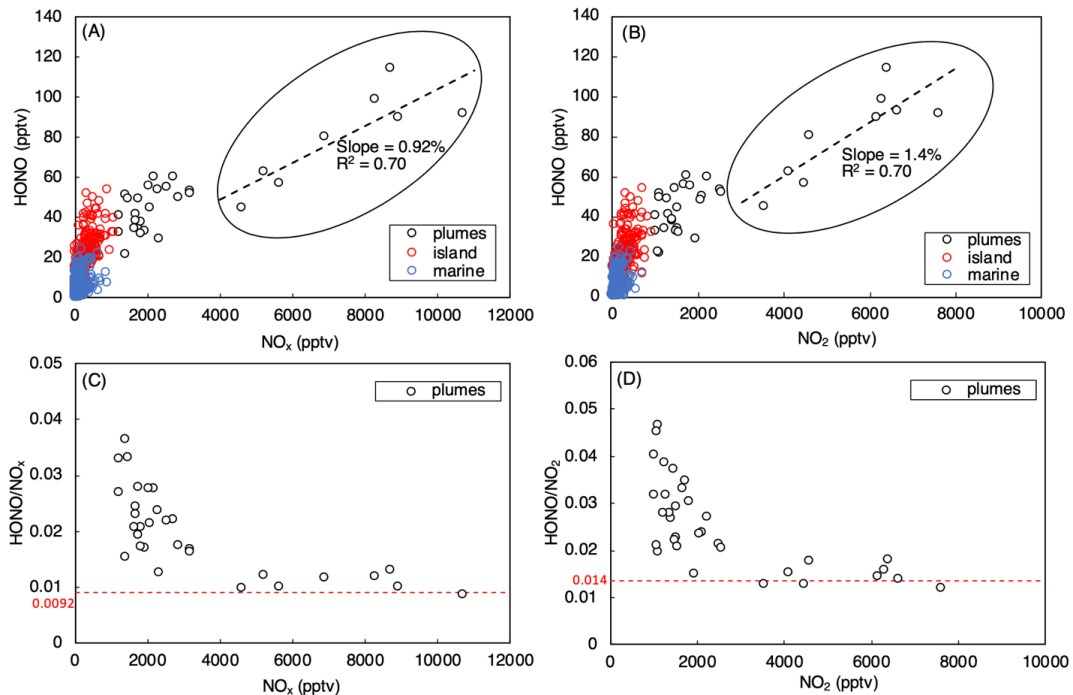

**Figure 5:** HONO concentrations plotted against $NO_x$ concentration and $NO_2$ concentration during daytime from 10:00 to 15:00. Data were divided into three categories (i.e., polluted plumes, island-influenced air, and clean marine air) based on local wind direction and $NO_2$ concentrations. The black dash lines in panels A and B represent the best fits of linear regression between HONO and $NO_x$ and between HONO and $NO_2$, respectively, in polluted plumes ($0 < $ wind direction $< 180°$, $[NO_2] > 3$ ppbv). The red dashed lines in panels C and D indicate where HONO/$NO_x$ = 0.0092 and HONO/$NO_2$ = 0.014, respectively.

Figs. 5A and 5B indicated that HONO concentration increased with $[NO_x]$ and $[NO_2]$, as expected from NO and $NO_2$ being HONO precursors. It was also shown that the HONO/$NO_x$ and HONO/$NO_2$ ratios decreased with $[NO_x]$ and $[NO_2]$ (Figs. 5C and 5D) due to the decreasing relative contribution from other HONO-formation mechanisms. With the increase in $NO_x$ and $NO_2$ concentrations in the pollution plumes, the $NO_x$-related processes became the dominant HONO precursor, leading to a stabilization in the HONO/$NO_x$ and HONO/$NO_2$ ratios at $[NO_2] > 3$ ppbv (Figs. 5C and 5D). In the daytime plumes ($[NO_2] > 3$ ppbv, 10:00–15:00), HONO concentrations were found to be strongly correlated with both $[NO_x]$ (Slope = 0.0092, $R^2 = 0.70$, Fig. 5A) and $[NO_2]$ (Slope = 0.014, $R^2 = 0.70$, Fig. 5B), as highlighted within black circles in Fig. 5. The slopes of [HONO]-$[NO_x]$ and [HONO]-$[NO_2]$ plots (i.e, 0.0092 and 0.014 respectively), agreed well with the stabilized HONO/$NO_x$ and HONO/$NO_2$ ratios displayed in Figs. 5C and 5D. Constant HONO/$NO_x$ and HONO/$NO_2$ ratios



of 0.0092 and 0.014 were then used to estimate the efficiency of HONO production through NO$_x$-related reactions:

$$P_{NO_x \rightarrow HONO} = 0.0092 \times [NO_x] \times J_{HONO} \qquad (5)$$

$$P_{NO_x \rightarrow HONO} = 0.014 \times [NO_2] \times J_{HONO} \qquad (5')$$


The calculations for $P_{NO_x \rightarrow HONO}$ in equations (5) and (5') were extrapolated to different air types categorized by local influences, assuming that the chemical kinetics of HONO production through NO$_x$-related processes were independent on NO$_x$ concentration levels. The median daytime $P_{NO_x \rightarrow HONO}$ values in clean marine air and island-

influenced air are $1.4 \times 10^{-3}$ pptv·s$^{-1}$ and $3.4 \times 10^{-3}$ pptv·s$^{-1}$, respectively if estimated by equation (5), and are $1.6 \times 10^{-3}$ pptv·s$^{-1}$ and $4.4 \times 10^{-3}$ pptv·s$^{-1}$, respectively if estimated by equation (5'). Since the data quality of LPAP NO$_2$ was significantly better than chemiluminescence NO$_x$ in the low-NO$_x$ marine and island-influenced air masses, the results derived from equation (5') were adopted (results are listed in Table 2). It should be noted that the calculated $P_{NO_x \rightarrow HONO}$ represents the upper limit estimate in clean marine air. Due to high solubility of

HONO in alkaline seawater, the ocean surface is likely a net HONO sink. Therefore, the HONO production efficiencies through NO$_x$-related processes were expected to be lower in the air mass in contact with the ocean surface than that over the island.

Data listed in Table 2 indicated that calculated $P_{NO_x \rightarrow HONO}$ contributed from NO$_x$-related processes failed to explain the HONO production budget needed to compensate the rapid HONO photolytic loss rates, $L_{photolysis}$.

The daytime missing HONO source was commonly reported by previous field studies in forested, marine, and urban environments (e.g., Elshorbany et al., 2009; Oswald et al., 2015; Su et al., 2011; Ye et al., 2016b), reflecting the source strength of HONO production from processes other than NO$_x$ reactions, such as pNO$_3$ and surface HNO$_3$ photolysis. $P_{missing}$ can be calculated based on the HONO budget analysis in equation (2):

$$P_{missing} = L_{photolysis} - P_{NO_x \rightarrow HONO} = P_{pNO_3 \rightarrow HONO} + P_{HNO_{3(ads)} \rightarrow HONO} + P_{other} \qquad (6)$$

The median $P_{missing}$ values of $2.9 \times 10^{-3}$ pptv·s$^{-1}$ and $1.9 \times 10^{-2}$ pptv·s$^{-1}$ accounted for 65% and 81% of the median HONO production rates needed to counter HONO photolytic loss rates in clean marine air and island-influenced air, respectively.

### 3.4.2 pNO$_3$ photolysis

In clean marine air where ground-surface related reactions are absent, pNO$_3$ might serve as an important HONO

precursor that explains its missing daytime source. Various research works, including those conducted in the field (Ye et al., 2016, 2018) and the laboratory (Ye et al., 2017b) reported significant enhancement in the photolysis rate constant of particulate nitrate ($J_{pNO_3}$) compared to that of gaseous nitric acid ($J_{HNO_3}$). To date, the largest dataset for $J_{pNO_3}^N$ was reported by Ye et al. (2017b) which included experimental results for aerosols collected from various environments, including "ground" samples from rural, urban and remote mountain areas,

as well as "aloft" samples from the troposphere above the Southeastern US. For clean marine environment, Ye





et al. (2016) reported 150–300 fold enhancement in pNO₃ photolysis relative to HNO₃ photolysis based on experimental data for a single aerosol sample collected on an aircraft during a research flight to the Atlantic Ocean off North and South Carolinas. However, it is possible that such high enhancement factor (EF) cannot be extrapolated to a larger geographical scale (e.g., the global oceanic environments), since several reactive
nitrogen chemistry models suggested that the better approximation for reactive nitrogen cycling is achieved when pNO₃ photolysis is assumed to be moderately enhanced (e.g., ~10–25 times higher) relative to the photolysis of HNO₃ (Kasibhatla et al., 2018; Reed et al., 2017; Romer et al., 2018; Ye et al., 2017b). In order to evaluate the contribution of pNO₃ photolysis on the daytime production of HONO in clean marine air of Bermuda, we assumed it accounting for 100% of the $P_{missing}$, and calculated an upper-limit enhancement factor
for pNO₃ photolysis relative to HNO₃ photolysis in clean marine air ($EF^*$):

$$EF^* = \frac{P_{missing}}{[NITs] \times J_{HNO_3}} \qquad (7)$$

where $J_{HNO_3}$ is the gaseous nitric acid photolysis rate constant and $[NITs]$ is the particulate nitrate concentration determined from the aerosol samples collected on Teflon filters. The median value and the 25th to 75th percentiles of the calculated $EF^*$ were 30, 18 and 54, respectively. The calculated median $EF^*$ value agrees with
several recent studies (Kasibhatla et al., 2018; Reed et al., 2017; Romer et al., 2018; Ye et al., 2017b) that pNO₃ photolysis is moderately enhanced relative to the photolysis of HNO₃ in the marine environments, but is much lower than the value between 150 and 300 reported by Ye et al. (2016). The discrepancies may represent the differences in the environments encountered in these different studies, i.e., the lower MBL in this work and other previous ground-based studies (e.g., Kasibhatla et al., 2018; Reed et al., 2017) *vs* the upper marine
boundary layer encountered in the aircraft study by Ye et al (2016). Significantly higher pNO₃ concentrations were observed in marine air at the THMAO site (medians of 284 and 120 pptv in spring and summer, respectively, Table 1) than in the upper marine boundary layer (~50 pptv) (Ye et al., 2016), with a much high pNO₃/HNO₃ ratio, ~ 6 in this work *vs* ~ 1 reported by Ye et al. (2016). Furthermore, super-micron particles were found to dominate the aerosol size distribution at the THMAO site, reflecting the younger age of marine
aerosols generated by breaking waves (Keene et al., 2007). Laboratory studies have shown that pNO₃ photolysis were enhanced by lower pNO₃ concentration, smaller aerosol particle size, and higher acidity, resulting in photolysis rate constants that varied by two orders of magnitude (Bao et al., 2018; Ye et al., 2017b).

The HONO production rates by pNO₃ photolysis assuming an $EF^*$ value of 30 ($P^*_{pNO_3 \rightarrow HONO}$) were estimated as:

$$P^*_{pNO_3 \rightarrow HONO} = 30 \times J_{HNO_3} \times [NITs] \qquad (8)$$

The medians of $P^*_{pNO_3 \rightarrow HONO}$ in clean marine air and island-influenced air are listed in Table 2. While the missing HONO production rate in the clean marine air masses can be accounted for by $P^*_{pNO_3 \rightarrow HONO}$, large imbalances in the HONO budgets still remain in the island-influenced air masses. That is, $P^*_{pNO_3 \rightarrow HONO}$ could only account for $2.7 \times 10^{-3}$ pptv·s⁻¹, less than 20% of the missing HONO production rate of $1.9 \times 10^{-2}$ pptv·s⁻¹ for



the island-modified air masses (Table 2), suggesting that there existed other mechanisms that were important in the formation of HONO during the day.

Laboratory-based photochemistry experiments were performed to determine pNO$_3$ photolysis rate constants using aerosol samples collected on Teflon filters during the field campaigns. The determined rate constant ($J_{pNO_3}^N$), normalized to the tropical noontime conditions at ground level with a $J_{HNO_3}$ of 7×10$^{-7}$ s$^{-1}$, varies from 1.01×10$^{-6}$ s$^{-1}$ to 1.81×10$^{-5}$ s$^{-1}$ with a mean value (±SD) of 3.19 (±2.33) ×10$^{-6}$ s$^{-1}$ and a median value of 2.56×10$^{-6}$ s$^{-1}$. The measured enhancement factor ($EF^m$) was calculated as:

$$EF^m = \frac{J_{pNO_3}^N}{7×10^{-7}} \tag{9}$$

The $EF^m$ value varies from 1.5 to 26, with a mean value (±SD) of 4.6 ± 3.3 and a median of 3.7. The median $EF^m$ value is significantly lower than the median $EF^*$ value of 30 derived from field observations. The low measured enhancement factor is similar to the recent reported EF values of ≤10 determined for NaNO$_3$ and HN$_4$NO$_3$ particles (Shi et al., 2021), but much lower than the values for ambient aerosol particles collected in filters from various environments (Bao et al., 2018; Ye et al., 2016, 2017b). While no categorial differences in $EF^m$ are observed for different types air masses under influences of island-modification or long-range atmospheric transport, the measured $EF^m$ value decreases with pNO$_3$ concentration, as previously reported (Ye et al., 2017b).  Furthermore, the highest $EF^m$ values are found to be associated with the fast-transported air masses from North America's eastern seaboard which represented high HNO$_3$/pNO$_3$ ratios, suggesting the enhancing effect of aerosol acidity (Bao et al., 2018). The large discrepancy between $EF^m$ and $EF^*$ might result from a bias in photolysis rate constants determined in the laboratory using stored marine aerosol samples collected on Teflon filters. The sampling and storage may alter the physical and chemical properties of the sea-salt aerosols, such as aggregation of particles and deprotonation of nitrate, resulting in the lower nitrate photo-reactivity compared to pNO$_3$ in the ambient aerosol particles in real marine atmospheric environment. Additional details for the temporal variations and potential factors affecting pNO$_3$ photolysis rate constants are to be discussed in a separate manuscript (Zhu et al., in preparation).

### 3.4.3 Nitric acid photolysis on island surfaces

The photolysis of nitric acid/nitrate adsorbed (HNO$_{3(ads)}$) on surfaces (e.g., the forest canopy), as suggested by previous field observations conducted in forested, rural area of North Michigan (Zhou et al., 2011) and Northeastern New York (Zhou et al., 2003, 2007), played a dominant role as HONO daytime source in low-NO$_x$ environment. In this field study, island-modified air masses may interact with the island surfaces, especially the forest surfaces that half surrounds the sampling site at Tudor Hill.  In the high-NO$_x$ plumes, NO$_x$ should be the dominant HONO precursor via reactions (R2) and (R3).  However, these high-NO$_x$ plumes picked up pollutants at point sources, such as power plant and cruise ships, and from small high traffic areas, such as city of Hamilton, and only accounted for small fraction of air masses modified by the island.  For the majority of the air masses, NO$_x$ concentration was relatively low (≤ 1 ppbv), and the photolysis of nitric acid/nitrate adsorbed (HNO$_{3(ads)}$) on island surfaces may be an important HONO source. To investigate the potential contribution from





HNO$_{3(ads)}$ photolysis on island surfaces, the unaccounted HONO production rate by NO$_x$-related processes and

pNO$_3$ photolysis ($P^*_{unaccounted}$) is calculated:

$$P^*_{unaccounted} = [HONO] \times J_{HONO} - 0.014 \times [NO_2] \times J_{HONO} - 30 \times J_{HNO_3} \times [NITs] \qquad (10)$$

The combined contribution from NO$_x$-related reactions and pNO$_3$ photolysis accounts for only ~7.1×10$^{-3}$ pptv s$^{-1}$ or ~30% of HONO production rate needed to compensate the HONO photolytic loss rate of ~2.4×10$^{-2}$ pptv s$^{-1}$ (Table 2). When the noontime $[HONO]_{unaccounted}$ (= $P^*_{unaccounted}/J_{HONO}$) is plotted against the averaged HNO$_3$

concentration over the prior 24 h ([HNO$_3$]$_{ave}$), a proxy for HNO$_{3(ads)}$ loading on the island surfaces, a significant correlation is found (R$^2$ = 0.54, Fig. 6), suggesting that HNO$_{3(ads)}$ is an important precursor of HONO during the daytime. The nitric acid photolysis on surface could potentially contribute to the majority of the missing HONO daytime source (i.e., ~ 1.6×10$^{-2}$ pptv s$^{-1}$) in island-influenced air masses (Table 2).

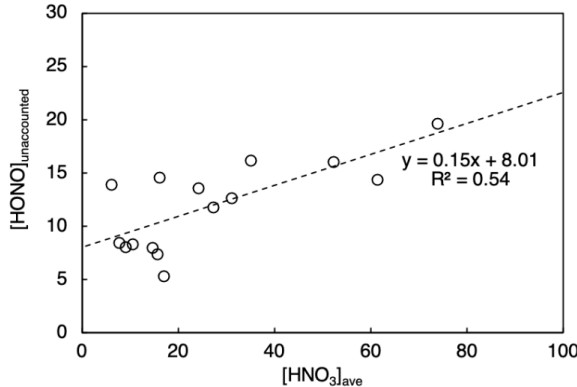


**Figure 6:** HONO concentration unaccounted by NO$_x$-related processes and pNO$_3$ photolysis ([HONO]$_{unaccounted}$, pptv) plotted against HNO$_3$ concentration averaged for 24 h prior to local noontime ([HNO$_3$]$_{ave}$, pptv) in island-influenced air with [NO$_2$] < 1 ppbv.






**Table 2:** Calculated HONO photolytic loss rate ($L_{photolysis}$), HONO production through $NO_x$-related reactions ($P_{NO_x \to HONO}$), missing HONO source ($P_{missing}$), estimated HONO production rate from the photolysis of $pNO_3$ ($P^*_{pNO_3 \to HONO}$) and the photolysis of $HNO_{3(ads)}$ ($P^*_{HNO_{3(ads)} \to HONO}$) in clean marine air and island-influenced air. The $P^*_{pNO_3 \to HONO}$ values were calculated assuming an $EF^*$ of 30. All the rates are in $10^{-3}$ pptv·s$^{-1}$. Measurement data from 10:00 to 15:00 were selected. Numbers listed in the table denote the medians and the values in the brackets represent $25^{th} - 75^{th}$ percentiles.

|  | $L_{photolysis}$ | $P_{NO_x \to HONO}$ | $P_{missing}$ | $P^*_{pNO_3 \to HONO}$ | $P^*_{HNO_{3(ads)} \to HONO}$ |
|---|---|---|---|---|---|
| Marine | 4.5 (2.6 – 8.0) | 1.6 (0.80 – 2.7) | 2.9 (1.8 – 5.3) | 2.9 | - |
| Island-influenced | 24 (18 – 34) | 4.4 (3.0 – 6.5) | 19 (15– 27) | 2.7 | 16 |

### 3.5 Case studies

Two case studies are presented here to show how the temporal variations in HONO can be caused by the various factors discussed in the previous sections:

Fig. 7 displayed a case study illustrating the temporal variations of HONO in response to various types of local influences during the period of May 10–12. From the midnight of May 10 to 18:00 on May 11, the site was dominated by local winds from the east and was thus under the island influence. The observed $NO_x$ levels were 570 sensitive to wind direction, higher in air masses from the northeastern quarter with a power plant, major roads, and population centers, and lower in air masses from southeastern quarters. In polluted plumes (highlighted in blue), average $NO_x$ concentration was 9 ppbv and the highest value reached 27 ppbv; the elevated $NO_x$ levels were associated with scavenging of $O_3$. The HONO concentration in these high-$NO_x$ plumes also reached the highest of 63 pptv in the selected period with relatively low HONO/$NO_2$ ratios (< 0.012), with $NO_x$-related 575 reactions (R2) and (R3) being the dominant HONO source. During the after-midnight hours (0:30–5:00) of May 11, the $NO_x$ level was low, with an average of 62 pptv, due to the lack of active emission sources. However, the HONO level (5–15 pptv) and HONO/$NO_2$ ratio (~0.10) were higher than their typical nighttime values in the clean marine air, probably due to the delayed release of HONO from heterogeneous $NO_2$ reaction occurred on island surfaces during high-$NO_2$ period a few hours earlier. From 18:00 on May 11 to the end of the selected 580 period, the local wind shifted to the southwesterly and westerly, and the site started receiving marine air without modification by the island, with low HONO and $NO_x$ levels most of the time. There was a short period (highlighted in pink) showing elevated $NO_x$ attributed to ship emission but no HONO spike, suggesting the



NO$_x$-related reactions was not a major HONO source in the marine air without interacting with the island, likely due to the limited surface area for heterogeneous NO$_2$ reactions in air masses over the ocean. The noontime

HONO/NO$_2$ ratios (0.04 on May 12) that were higher than those in polluted plumes (<0.012) support the argument that the photolysis of pNO$_3$ (R5) was the main contributor to the daytime formation of HONO in clean marine air (Table 2). The highest HONO/NO$_2$ ratios (~0.13) were observed near the solar noon of the island-influenced period from May 10–11, reflecting active photochemical formation of HONO contributed via multiple mechanisms, including homogeneous and heterogeneous NO$_x$ reactions (R2) and (R3), and the

photolysis pNO$_3$ associated with aerosols (R5) and HNO$_3$ on the island surfaces (R4).

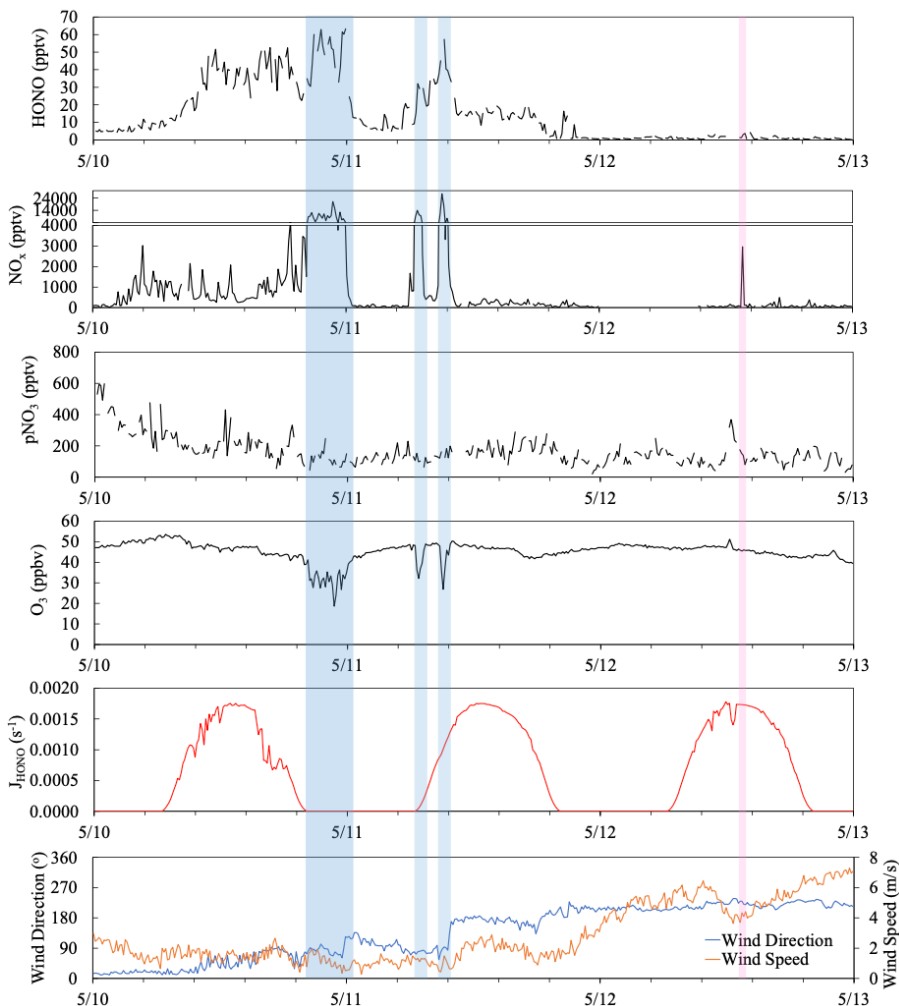

**Figure 7:** Time series plots of HONO, NO$_x$, pNO$_3$, O$_3$, HONO photolysis rate constant (J$_{HONO}$), wind direction
and wind speed from May 10 to 12, 2019. The light blue boxes highlighted highly polluted periods when the
Tudor Hill site received urban pollutants. The pink box highlighted a short-term pollution plume resulted from
ship emissions.

Fig. 8 depicted a case study showing the impact of long-range air mass transport on the temporal distribution of
reactive nitrogen species. During the selected period from September 7–9, the site was dominated by the
westerly flows (180º < wind direction < 360º) and thus the sampling site received mostly the clean marine air.
There was only one short period near 8:00 on September 8 (highlighted in yellow) showing elevated HONO and
NO$_x$ concentrations attributed to the local emissions and island surface effect when local wind direction shifted


briefly to the southeasterly. There were also a few spikes in $NO_x$ emitted from ships in the upwind sea-lanes on
September 8 and 9, but no HONO spikes due to the low surface area available for heterogeneous $NO_2$ reactions
in this air mass traveling over the ocean. During the summer field campaign, the air mass traveled to the
THMAO site was dominated by well-aged marine air above the Atlantic Ocean. Starting from the early
afternoon of September 7, hurricane Dorian disturbed the Bermuda high when it traveled along the eastern coast
of North America; the modeled backward trajectories indicate that the air mass arriving at the site during the
period originated from the Northeastern US. The fast-traveling air flow from North America contained elevated
levels of particulate nitrate (219– 585 pptv) and $O_3$ (29 – 49 ppbv) than those in aged marine air (average
concentration for $pNO_3$ and $O_3$ were 127 pptv and 16 ppbv, respectively). During the air mass transition, we
observed a strong increasing trend in HONO and $HNO_3$ concentrations from the evening of September 7 to the
early morning of September 8. The rapid increase in $HNO_3$ concentrations (from ~7 to 110 pptv) during a 24-h
period from 8:00 on September 7 to 8:00 on September 8 largely affected the partitioning of nitrate in the gas
phase and the aerosol phase ($HNO_3$/NITs increased from 4% to 25%). An elevated daytime maximum in HONO
concentration was found on September 8 (~6 pptv, three times higher than the ~2 pptv daytime HONO
maximum on September 7). The photoproduction efficiency of HONO from aerosols also changed; the average
value of $J_{pNO_3}^N$ determined for aerosol samples collected on September 8 was $1.0\times10^{-5}\,s^{-1}$, much higher than the
summer average value of $J_{pNO_3}^N$ (i.e., $3.4\times10^{-6}\,s^{-1}$). The HONO production efficiency from $pNO_3$ photolysis
appears to be enhanced by the acidity of the aerosols.

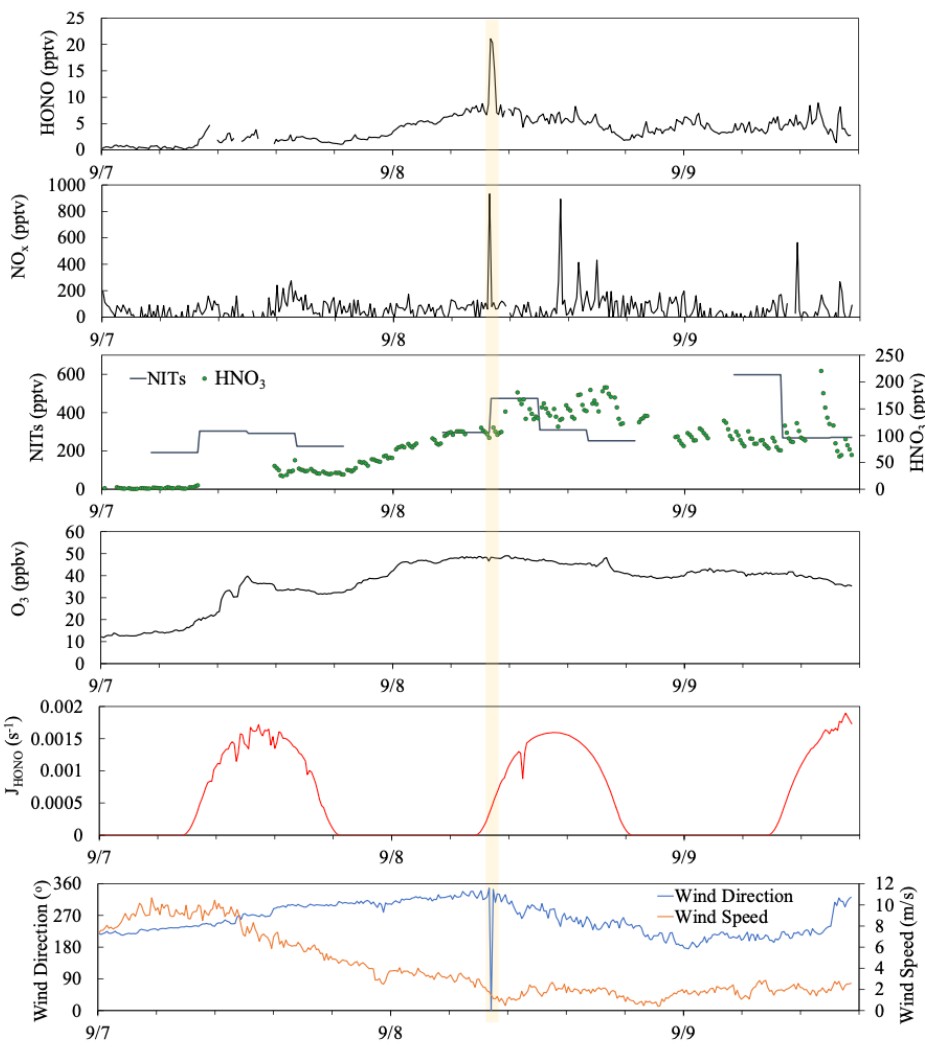

**Figure 8:** Time series plots of HONO, $NO_x$, NITs, $HNO_3$, $O_3$, HONO photolysis rate constant ($J_{HONO}$), wind
direction and wind speed from September 7 to 9, 2019. The yellow box highlighted short-term pollution plume
resulted from a sudden shift in local wind direction.



## 4 Conclusions

Large temporal variations in HONO concentrations were observed at THMAO site in Bermuda during the spring and the summer of 2019, depending on the types of marine air masses and how the air masses interacting with the island. High concentrations of HONO, up to 278 pptv, were found in the locally emitted high-$NO_x$ plumes. A $HONO/NO_2$ ratio of 0.014 is derived from the HONO-$NO_2$ correlation in the daytime plumes and is used to estimate the HONO source strength from $NO_x$-related reactions during the day. HONO concentrations were in low pptv and ten pptv levels in the low-$NO_x$ marine air masses without and with island modification, respectively. Distinctive diurnal HONO variations were observed in both low-$NO_x$ marine air with or without island modification, with daytime plateaus and nighttime valleys. Such diurnal variation patterns suggest that HONO was produced by photochemical processes during the day and that ocean surface was likely a net HONO sink. The photolysis of $pNO_3$, if moderately enhanced relative to $HNO_3$ photolysis by a factor of 30, can explain the missing daytime HONO source in the clean marine air mass, which account for the majority (~65%) of the overall daytime HONO budget of $\sim 4.5 \times 10^{-3}$ pptv s$^{-1}$. To sustain the observed daytime HONO concentration in the low-$NO_x$ island-modified air mass, a large HONO source, $\sim 1.6 \times 10^{-2}$ pptv s$^{-1}$, was required, in addition to the known HONO production sources from $NO_x$ and $pNO_3$ precursors, $4.4 \times 10^{-3}$ pptv s$^{-1}$ and $2.7 \times 10^{-3}$ pptv s$^{-1}$, respectively. The photolysis of $HNO_{3(ads)}$ on forest canopy and other island surfaces was likely responsible for this large missing HONO source in the low-$NO_x$ island-modified air mass. The higher concentrations in HONO and its precursors, $NO_x$ and $pNO_3$, were higher in spring than in summer in the clean marine air masses, attributed to the fact that the air masses arriving at the site in the summer had circulated over the North Atlantic Ocean and were more aged than air masses that traveled from the North American continent in the spring. The "background" concentrations of HONO precursors decreased as the air masses aged and thereby lowered the production rates of HONO.

## 5 Author Contribution

XZ and YE designed the study. YZ, YW and XZ performed the development, calibration, and deployments of the reactive nitrogen measurements systems. XZ led the field campaigns. All authors contributed to the data collections during the field campaigns. YZ conducted the photochemistry experiments, performed the data processing, and prepared the manuscript with inputs from all coauthors.

## 6 Competing Interests

The authors declare no conflicts of interests.

## 7 Acknowledgment

This research is supported by the National Science Foundation's Atmospheric Chemistry Program and Chemical Oceanography Program through the grants AGS-1826989 (Xianliang Zhou), AGS-1826956 (Yasin Elshorbany), OCE-1829686 (Andrew Peters), and by Chinese National Natural Science Foundation through the grant 41875151 (Chunxiang Ye). We thank Dr. Haider Khwaja and lab members at Wadsworth Center for providing instrumentation and laboratory supports.



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
