# Peer review of "An investigation into the chemistry of HONO in the marine boundary layer at Tudor Hill Marine Atmospheric Observatory in Bermuda"

_Atmospheric Chemistry and Physics, 2021_

## Author Comment (AC1)

**Response to Reviewers**

**An investigation into the chemistry of HONO in the marine boundary layer at Tudor Hill Marine Atmospheric Observatory in Bermuda**

by Yuting Zhu and Xianliang Zhou

**Reviewer #1**

In the manuscript by Zhu et al., the chemistry of HONO at a marine measurement site on Bermuda island was investigated. Different HONO sources were discussed for which $NO_x$ related reactions were considered important under polluted island influenced conditions, whereas HONO formation by particle nitrate photolysis was found to be more important under clean marine conditions. Furthermore, photolysis of ground surface adsorbed $HNO_3$ was postulated as main HONO source reaction for low $NO_x$ island influenced conditions. An important observation is the missing night-time formation of HONO under clean marine conditions, which is reasonable considering the alkaline sea surfaces acting as a perfect sink for HONO. This result agrees with another recent paper (Crilley et al., ACP, 2021, not cited), but is in contrast to former observations from China and Canada.

I have several comments, which should be considered in the revised paper.

We sincerely thank the reviewer for the detailed comments and the helpful suggestions, which help improving our manuscript. Our responses to the reviewer's comments are colored in blue below.

We appreciate that the reviewer pointed out that Crilley et al. (2021) was not cited in the original manuscript. This recent paper is now cited in the revised manuscript.

**Major Comments:**

1) Description of the HONO daytime budget:

For the production rate (equation 2) $NO_x$ related HONO formation is not well described. First, HONO formation by the gas phase reaction NO+OH could be easily implemented by assuming a reasonable diurnal OH profile (e.g. by its correlation with $J(O^1D)$, or by any simple box model). Since the homogeneous formation is most probably not too important here, even large uncertainties (e.g. factor of two…) will not matter too much. When OH is calculated, also the (minor) loss of HONO by its reaction with OH could be explicitly considered besides the HONO photolysis (see equation 4, where this reaction is now neglected). Considering OH, the PSS of HONO can be calculated and only excess levels (and not measured HONO…) should be explained by the discussed sources ($P_{extra}$).

Second and much more important, HONO formation by $NO_2$ conversion during daytime must be implemented as a photolytic process, which is typically parameterized as a function of the $NO_2$ concentration and (!) $J(NO_2)$, see work by Stemmler et al.. In this context, also reaction (R3) must include its photosensitized character! Please add hv to the reaction (compare R4). A dark reaction R3 is also known, but since night-time HONO formation was not observed, this is not of importance here. When the photosensitized conversion is correctly included, most probably the diurnal shape of the HONO production ($P_{extra}$) will be well explained (see below). Thus, the different patterns of HONO (one daytime maximum) and $NO_2$ (two maxima) (see lines 346-348 and Fig. 4) cannot demonstrate the missing importance of reaction (R3) as long as this reaction is not correctly considered.

We observed a strong correlation between the noontime concentrations of HONO and $NO_x$ under high-$NO_x$ conditions in this study. The observed HONO-$NO_x$ relationships were extrapolated to low-$NO_x$ conditions in the original manuscript for a simplified estimates for HONO production rates from $NO_x$-related reactions. We agree with the reviewer that this simplified approach has significant limitations due to the non-linear relationship between the photosensitized reaction rate and $NO_2$ concentration, as well as changes in air mass types and conditions (e.g., with or without contact with ground surfaces).

We follow the reviewer's suggestions and estimate daytime HONO production rates from gas-phase NO+OH reaction and $NO_2$ heterogeneous reaction separately in the revised manuscript, in the following HONO budget equation:

$$\frac{d[HONO]}{dt} \approx 0$$

$$= P_{NO+OH \to HONO} + P_{NO_{2(aerosol)} \to HONO} + P_{NO_{2(ground)} \to HONO} + P_{pNO3 \to HONO} +$$
$$P_{HNO3_{(ads)} \to HONO} + P_{other} - (L_{photolysis} + L_{HONO+OH} + L_{deposition}) \qquad \text{(Re1)}$$

For clean marine air, the following equations are used to estimate the HONO production rates through R2 and R3 in the original manuscript:

$$P_{NO+OH \to HONO} = k_{NO+OH} \times [OH] \times [NO] \qquad \text{(Re2)}$$
$$P_{NO_{2(aerosol)} \to HONO} = k_{NO_2-aerosol} \times [NO_2] \qquad \text{(Re3)}$$

Where $k_{NO+OH}$ is the reaction rate constant between NO and OH obtained from Atkinson et al. (2004), and a constant [OH] of $6 \times 10^6$ molecules cm$^{-3}$ is assumed. For equation Re3, $k_{NO_2-aerosol}$ is calculated using the following equation:

$$k_{NO_2-aerosol} = \frac{1}{4} \times \overline{v_{NO_2}} \times \frac{S}{V} \times \gamma_{NO_2-aerosol} \qquad \text{(Re4)}$$

Where $\overline{v_{NO_2}}$ is the average molecular speed of $NO_2$, $\frac{S}{V}$ is the surface area to volume ratio, and $\gamma_{aerosol}$ is the uptake coefficient of $NO_2$ on aerosol surfaces. A $\frac{S}{V}$ ratio of $5 \times 10^{-5}$ m$^{-1}$ is used, based on 20 µg·m$^{-3}$ of 1-µm sea-salt aerosol particles. An upper limit $\gamma_{NO_2-aerosol}$ value of $2 \times 10^{-5}$ is also used, taking account the photo-enhancement of HONO formation through heterogeneous reaction of $NO_2$ (Li et al., 2010; Stemmler et al., 2006). The calculated

medians for $P_{NO+OH \rightarrow HONO}$ and $P_{NO_2(aerosol) \rightarrow HONO}$ are $8.9 \times 10^{-4}$ and $5.9 \times 10^{-6}$ pptv·s$^{-1}$ in clean marine air, which only account for minor fractions (21% and 0.14%, respectively) of the median HONO production rate needed to sustain HONO photolytic loss. A large fraction (~79%) of HONO production rate cannot be accounted for by the contribution from NO$_x$-related reactions. $P_{NOx \rightarrow HONO}$, defined here as the sum of $P_{NO+OH \rightarrow HONO}$ and $P_{NO_2(aerosol) \rightarrow HONO}$ is estimated to be $9.0 \times 10^{-4}$ pptv·s$^{-1}$ in clean marine air, is lower than our previous estimate of $1.4 \times 10^{-3}$ pptv·s$^{-1}$ by equation (5') in the original manuscript.

We also agree with the reviewer that the daytime production of HONO from NO$_2$ heterogeneous reaction is more important as a HONO source in island-influenced air than clean marine air. We calculate the HONO concentration that cannot be explained by gaseous production ([HONO]$_{unexplained}$):

$$[HONO]_{unexplained} = [HONO] - \frac{P_{NO+OH \rightarrow HONO}}{J_{HONO} + k_{HONO+OH} \times [OH] + \frac{\upsilon_{HONO}}{H}} \tag{Re5}$$

Where $J_{HONO}$ is the HONO photolysis rate constant, $k_{HONO+OH}$ is the HONO+OH reaction rate constant, $\upsilon_{HONO}$ is the deposition velocity, which is set to a high value of 1 cm s$^{-1}$ (Harrison et al., 1996; Stutz et al., 2002), and $H$ the vertical transport distance. The value of $H$ (i.e., 116 m) was calculated following Zhang et al. (2009), with an assumed turbulent diffusion coefficient K$_z$ of $10^5$ cm$^2$ s$^{-1}$ and a HONO photolytic lifetime of 670 s (11 min). [HONO]$_{unexplained}$ is then plotted against [NO$_2$] in island-influenced air (Panel A) and clean marine air (Panel B) in Fig. Re1. A slight correlation was found for island-influenced air (R$^2$=0.23), and no correlation was observed for clean marine air (R$^2$=0.03), supporting the arguments that heterogeneous NO$_2$ reaction contribute to a certain fraction of observed HONO concentration in the island-influenced air, but not in the clean marine air.

[Figure]

**Figure Re1:** [HONO]$_{unexplained}$ plotted against [NO$_2$] in (A) island influenced air and (B) clean marine air from 10:00 to 15:00. The dash lines indicate the best-fit lines for linear regression between the two parameters in each figure.

We use these following equations to estimate the rate and rate constant for NO$_2$ to HONO conversions on ground surfaces ($P_{NO_{2(gound)} \rightarrow HONO}$):

$$P_{NO_{2(gound)} \rightarrow HONO} = k_{NO_2-ground} \times [NO_2] \qquad \text{(Re6)}$$

$$k_{NO_2-ground} = \frac{1}{4} \times \overline{v_{NO_2}} \times \frac{S}{V} \times \gamma_{NO_2-ground} \qquad \text{(Re7)}$$

An S/V value of 0.017 m$^{-1}$ was calculated with an effective surface area of 2 m$^2$ per geometric surface area and an air column height ($H$) of 116 m. The value of $H$ was calculated following Zhang et al. (2009) assuming a turbulent diffusion coefficient K$_z$ of 10$^5$ cm$^2$ s$^{-1}$, and a HONO photolytic lifetime of 670 s (11 min). $\gamma_{NO_2-ground}$ is set as 2×10$^{-5}$, which is a upper-limit value suggested by Stemmler et al. (2006). The calculated median of $P_{NO_2-ground}$ in island-influenced air is 7.0×10$^{-3}$ pptv·s$^{-1}$ and accounts for 28% of the HONO daytime production budget in island-influenced air masses.

We would like to thank the reviewer for pointing out that $\gamma_{NO_2}$ is expected to decrease with $NO_2$ concentration. For this work, $NO_2$ heterogeneous reactions are only parameterized for low-$NO_x$ conditions ($[NO_2] < 1$ ppbv). We choose to use the upper-limit $\gamma_{NO_2}$ value of $2\times10^{-5}$ suggested by Stemmler et al. (2006) because our $NO_2$ level is at the lower end of the $NO_2$ concentration range in Stemmler et al. (2006), and significant decrease in $\gamma_{NO_2}$ is not expected when $NO_2$ concentration is below 1 ppbv.

The calculation results above indicate that heterogeneous production of HONO from $NO_2$ is more important as a HONO source in island-influenced air than clean marine air, as suggested by the reviewer. However, it should be noted that the combined HONO production from $NO_x$-related reactions is only a minor HONO source even in island-influenced air; 62% of HONO production rate is still missing after counting the contributions from $P_{NO+OH \rightarrow HONO}$, $P_{NO_{2(aerosol)} \rightarrow HONO}$, and $P_{NO_{2(gound)} \rightarrow HONO}$.

2) Proposed extra sources:

To identify potential HONO formation mechanisms during daytime, only the extra HONO sources (besides the known reaction NO+OH, and the know losses by photolysis and reaction with OH, see above) should be calculated and the diurnal profile of $P_{extra}$ be compared with different postulated sources, e.g. by plotting against different parameters, like J(NO₂)xNO₂, J(HNO₃)xHNO₃(av), J(HNO₃)xpNO3 etc. for clean marine and island influenced conditions. I am quite sure that the best correlation will be obtained for J(NO₂)xNO₂ for island influenced conditions (see below). This will be in contrast to the statement made simply from the concentration (…) profiles (see lines 346-348), which does not include the photolytic nature of the formation process by $NO_2$ reactions. In addition, there is always a non-linear connection between the HONO production and its concentration, caused by the variable main loss term (photolysis). In addition, reaction R3 is underestimated and R4 overestimated, see below.

HONO daytime production rate that cannot be explained by the gaseous reaction between NO and OH ($P_{unexplained}$) can be calculated as:

$$P_{unexplained} = [HONO] \times (J_{HONO} + k_{HONO+OH} \times [OH] + \frac{v_{HONO}}{H}) - P_{NO+OH \rightarrow HONO}$$

$$(Re8)$$

Here we follow the reviewer's suggestion and plot $[NO_2]\times J_{NO2}$, $[pNO_3]\times EF^*\times J_{HNO3}$, and $[HNO_3]_{ave}\times J_{HNO3}$ versus $P_{unexplained}$ in island-influenced air (Fig. Re2). $[HNO_3]_{ave}\times J_{HNO3}$ exhibit better correlation ($R^2 = 0.34$) with $P_{unexplained}$ than $[NO_2]\times J_{NO2}$ ($R^2 = 0.27$) and much better than $[pNO_3]\times EF^*\times J_{HNO3}$ ($R^2 = 0.16$).

[Figure]

**Figure Re2:** $[NO_2] \times J(NO_2)$, $[pNO_3] \times EF^* \times J(HNO_3)$, and $[HNO3]_{ave} \times J(HNO_3)$ plotted against $P_{unexplained}$ in island-influenced air. The dash lines indicate the best fit lines for linear regressions between the two parameters in each plot.

It is difficult to conclude whether one precursor is more important than the other by just comparing the $R^2$ values displayed in Fig. Re2, especially since the correlations are moderate at best. Therefore, in the revised manuscript, we calculate HONO production rate from gaseous reaction between NO and OH via equation (Re2), HONO productions from $NO_2$ heterogeneous reaction on aerosol surface ($P_{NO_{2(aerosol)} \rightarrow HONO}$) via equation (Re3) and on ground surface ($P_{NO_{2(ground)} \rightarrow HONO}$) via equation Re6, and the upper-limit HONO production rate via $pNO_3$ photolysis ($P^*_{pNO_3 \rightarrow HONO}$) using the following equation:

$$P^*_{pNO_3 \rightarrow HONO} = EF^* \times J_{HNO_3} \times [pNO_3^{filter}] \tag{Re9}$$

$EF^*$ is the upper-limit enhancement factor for pNO₃ photolysis obtained by attributing all the missing HONO production to pNO₃ photolysis in clean marine air. We add a new figure (Fig. Re3 in this response) in the revised manuscript to illustrate the contributions from different production and loss pathways to the HONO budget in island-influenced air. The loss terms were calculated using the following equation:

$$L_{HONO} = L_{photolysis} + L_{HONO-OH} + L_{deposition}$$
$$= [HONO] \times J_{HONO} + k_{HONO-OH} \times [HONO] \times [OH] + \frac{v_{HONO}}{H} \times [HONO] \tag{Re10}$$

It should be noted that $P_{NO_{2(aerosol)} \rightarrow HONO}$ contribute to less than <0.1% of the HONO production budget, and therefore is not displayed in Fig. Re3.

[Figure]

**Figure Re3:** Median values for the production and loss rates of HONO contributed by different processes in island-influenced air masses from 10:00 to 15:00.

As the reviewer pointed out, HONO production via HNO₃₍ads₎ photolysis ($P_{HNO_{3(ads)} \rightarrow HONO}$) in the original manuscript could be overestimated, since reaction R4 was assumed to account for 100% the remaining HONO production rate after removing the contributions from NOₓ-related reactions and pNO₃ photolysis. In the revised manuscript, we use the term "P_HNO₃₍ads₎→HONO + P_other" for $P_{unaccounted}$, which is defined as:

$$P_{unaccounted} = L_{HONO} - P_{NO+OH \rightarrow HONO} - P_{NO_{2(aerosol)} \rightarrow HONO}$$

$$-P_{NO_{2(ground)} \rightarrow HONO} - P^*_{pNO_3 \rightarrow HONO} \qquad \text{(Re11)}$$

The median value for $P_{unaccounted}$ in the revised manuscript is $1.3 \times 10^{-2}$ pptv·s$^{-1}$ for island-influenced air.

Reasons for a major HONO formation by photosensitized conversion of $NO_2$ on ground surfaces become obvious from Fig. 4:

- There are clear diurnal profiles for HONO and HONO/$NO_2$ with near noon maxima, pointing to a photochemical nature of the HONO sources;

- HONO levels for clean marine air masses (A) are factors of 4/7 (summer/spring) lower than for island influenced air masses (B), pointing to a major formation on island surfaces;

- Despite the large HONO differences for (A) and (B), pNO3 is similar for both air masses (compare pNO3 for (A) and (B)). Thus, any major HONO formation by particle nitrate photolysis is unlikely, at least for island influenced conditions.

- HONO is similar for spring and summer for clean marine conditions, although the $NO_2$ levels are very different (compare upper two figures for case (A)). Thus, heterogeneous HONO formation by any $NO_2$ reactions on ocean surfaces are most probably negligible. In contrast, for island influenced conditions (B), for which active surfaces for reaction (R3) are present (e.g. humic acid surfaces), HONO and $NO_2$ are both ca. a factor of two higher in spring than in summer.

All together these observations clearly indicate that HONO is mainly formed by photosensitized conversion of $NO_2$ on island surfaces.

We agree with the reviewer on all the arguments that (1) the clear diurnal profiles for HONO and HONO/$NO_2$ with near noon maxima point to a photochemical nature of the HONO sources; (2) much lower HONO levels in clean marine air than in island influenced air point to a major formation on island surfaces; (3) particulate nitrate photolysis is unlikely to be a major HONO source in island influenced air; and (4) HONO formation from heterogeneous $NO_2$ reaction is negligible in clean marine air but can be important in island influenced air. We have presented these arguments throughout our original manuscript. Please see our response to comment (1) and (2) for detailed discussions.

For island-influenced air under low-$NO_x$ conditions, both $NO_x$ and pNO3 were higher in the spring than the summer by a factor of ~2, leading to higher HONO concentration in the spring. Our budget analyses show that the median of $P_{NO_{2(ground)} \rightarrow HONO}$ only contribute to 28% of the total HONO production budget during the day, even with an upper-limit $\gamma_{NO_2}$ of $2 \times 10^{-5}$ used in equation Re7.

3) Parameterization of the $NO_x$ related reactions:

Besides the separation of HONO formation by gas phase reaction of NO+OH and by photosensitized reaction R3 (see above), for the latter reaction a non-linear correlation of the uptake coefficient with the $NO_2$ concentration is well known (decreasing gammas with increasing $NO_2$, see the cited papers by Stemmler et al.). Thus, besides that HONO production and not simply its concentration should be considered (see above), the non-linear correlation between the HONO and the $NO_2$ concentrations in Figure 5 is also not well described in the paper. Here the authors concluded a perfect first order kinetics of R3 (gamma independent of the NO2 concentration) and used only the low conversion efficiency, observed in the high $NO_x$ plumes, which will strongly underestimate this source for the low $NO_x$ island influenced conditions, later discussed. In addition, they simply used this lower limit HONO/$NO_x$ ratio and constantly applied it to parameterize the HONO production, despite its variable photochemical nature (see above). In conclusion, the used parameterization strongly underestimates the photosensitized $NO_2$ conversion for low $NO_x$ conditions.

According to our updated budget analyses, HONO daytime production contribution from $NO_2$ heterogeneous conversion is unimportant in clean marine air, and is marginally important in island-influenced air, accounting for ~0.14% and ~28% of the total daytime HONO production budget in clean marine air and island-influenced air, respectively. Please see our response to major comment (1) and (2) for details of the updated budget analyses.

4) Parameterization of HONO formation by pNO3 photolysis

In contrast to R3 the authors overestimated R4 by pNO3 photolysis. Here by equation (7), an upper limit EF* of 30 is determined by assuming that all missing HONO sources besides R3 are by pNO3 photolysis. First, if a higher contribution of R3 were considered (see above), EF* would get lower, bringing that closer to the lab values. Second, by using aerosol samples the authors determined a much more reasonable EF(m) of ca. 4 in the laboratory, which is in good agreement with other recent studies (Romer et al., 2018: "most reasonable range EF=1-7"; Shi et al., 2021: EF = ca. 1). I encourage the authors to use their own measured EF(m), making that source much less important, in agreement with the above discussed general observations in Fig. 4.

We have calculated HONO formation rates from gaseous NO+OH reaction (R2) and heterogeneous $NO_2$ reaction (R3), as suggested by the reviewer (see the discussions above). However, the contribution from $NO_x$-related reactions to the total daytime HONO production in clean marine air is actually lower in the updated budget analyses (21%, mostly from gaseous NO+OH reaction) than our original estimate (34%) extrapolated from HONO production efficiency in high-$NO_x$ plumes. The decreased contribution is expected since the HONO production efficiencies through $NO_x$-related processes are expected to be lower in the air mass in contact with the ocean surface than that over the island.

In the revised manuscript, we present both $EF^*$ calculated from field observation data and $EF^m$ obtained directly from laboratory measurements. We modify the discussion and add a figure (Fig. Re4) to compare our estimated HONO production with equation Re9 and equation Re12 below:

$$P^m_{pNO_3 \rightarrow HONO} = EF^m \times J_{HNO_3} \times [pNO_3^{filter}] \qquad \text{(Re12)}$$

[Figure]

**Figure Re4:** Median values for the production and loss rates of HONO contributed by different processes in clean marine air from 10:00 to 15:00. For HONO production rate via pNO3 photolysis, $P^*_{pNO_3 \rightarrow HONO}$ in panel (A) was estimated with a constant $EF^*$ value of 29 and $P^m_{pNO_3 \rightarrow HONO}$ in panel (B) was estimated with the measured enhancement factor $EF^m$.

The added Fig. Re4 delivers a clear message to the reader that there exists a large difference in our estimates for HONO production via pNO3 photolysis using $EF^*$ and $EF^m$. We also revised the text to point out to the reader that EF* is an upper-limit enhancement factor for pNO3 photolysis relative to HNO3 photolysis.

5) Proposed HONO formation by photolysis of adsorbed HNO3 on island surfaces

By the same argument, also the proposed major HONO formation by photolysis of ground surface adsorbed HNO3 will get lower, if the higher contribution of the NO2 reaction R3 is considered. Also, the high intercept in Figure 6 (independent of HNO3(av)…) compared to the HNO3(av) dependent HONO, is a strong argument why this ground surface photolysis of HNO3 is not mainly responsible for the missing HONO sources for island influenced conditions.

We followed the reviewer's suggestions and updated our budget analyses for the daytime chemistry of HONO, as discussed above. For island-influenced air, the sum of $P_{NO+OH \rightarrow HONO}$, $P_{NO_2(aerosol) \rightarrow HONO}$, and $P_{NO_2(ground) \rightarrow HONO}$, accounts for 38% of the total HONO production, higher than the 18% reported for $P_{NO_x \rightarrow HONO}$ in the original manuscript.

Extensive changes are made in the revised manuscript. The noontime HONO concentration that cannot be explained by $NO_x$-related reactions and $pNO_3$ photolysis ($[HONO]_{unaccounted}$) is defined as:

$$[HONO]_{unaccounted} = \frac{P_{unaccounted}}{J_{HONO} + k_{HONO-OH} \times [OH] + \frac{v_{HONO}}{H}} \qquad \text{(Re13)}$$

Please see to our reply to major comment (2) for the definition of $P_{unaccounted}$ (equation Re11). $[HONO]_{unaccounted}$ was found to be correlated ($R^2 = 0.50$, Fig. Re5) with the averaged $HNO_3$ concentration over the prior 24 h ($[HNO_3]_{ave}$, a proxy for $HNO_{3(ads)}$ loading on the island surfaces). We agree with the reviewer that the high intercept (i.e., ~5.6 pptv) in the $[HONO]_{unaccounted}$ versus $[HNO_3]_{ave}$ plot indicates that $HNO_3$ is unlikely responsible for 100% of $P_{unaccounted}$. Therefore, in the revised manuscript, we report $P_{unaccounted}$ as the sum of $P_{HNO_{3(ads)} \to HONO}$ and $P_{other}$. Please see Fig. Re3 and our response to comment (2) for additional details. Also, it should be noted that the intercept (Fig. Re5) might in part be attributed to the nonlinear correlation between $J_{HNO_{3(ads)} \to HONO}$ and the loading of $HNO_{3(ads)}$ on surfaces, which was reported in (Ye et al., 2016a).

[Figure]

**Figure Re5:** HONO concentration unaccounted by $NO_x$-related processes and $pNO_3$ photolysis ($[HONO]_{unaccounted}$, pptv, averaged for each day from 10:00 to 15:00) plotted against $HNO_3$ concentration averaged for 24 h prior to local noontime ($[HNO_3]_{ave}$, pptv) in island-influenced air with $[NO_2] < 1$ ppbv.

**Specific Comments:**

The following concerns are listed in the order how they appear in the manuscript.

Line 53, R3: either add a new, similar photosensitized reaction, or add "hv" to consider only daytime chemistry.

R3 is revised as:
$$NO_2 + H_2O, organics \xrightarrow{surface\ hv} \to HONO$$
in the revised manuscript

Line 62: The first study, which proposed significant extra daytime HONO production was by Neftel et al., 1996, please add.

Reference is added in the revised manuscript.

Line 72: Add a reference to the new photosensitized R3.

The references that support the enhanced HONO formation during photosensitized, heterogeneous reaction of $NO_2$ are now provided in the revised manuscript.

Lines 87-88: The very recent study by Crilley et al., 2021 (ACP) is missing.

Reference is added in the revised manuscript.

Page 5, $HNO_3$ and pNO3 measurement: It is not clear how uptake of nitrate containing particles in the stripping coil was considered for the $HNO_3$ detection and whether the wetted frit disc sampler measures the sum of pNO3 and $HNO_3$, as expected. How where these interferences considered?

The reviewer is correct that the coil sampler would collect a fraction of nitrate containing particles in the air stream though at lower efficiency ($\leq 20\%$) (Zhou et al., 2002) and that the wetted frit disc sampler could collect gaseous $HNO_3$ quantitatively. To correct for the potential $pNO_3$ interference in the $HNO_3$ measurement system, "zero-HONO/$HNO_3$" air was generated by pulling ambient air through a $Na_2CO_3$ denuder placed upstream to the coil sampler and was used to generate a baseline. Most of the interfering atmospheric constituents ($pNO_3$, $pNO_2$, $NO_x$, $pNO_2$, PAN) are expected to pass through the denuder and enter the coil sampler and their potential interference was eliminated by subtracting the baseline signal from the ambient signal when calculating ambient $HNO_3$ concentration.

In the $pNO_3$ measurement system, a $Na_2CO_3$ denuder was installed right before the wetted frit disc sampler to remove all the gaseous acidic species, including $HNO_3$ and HONO, and thus remove their potential interference. "Zero-$pNO_3$" air was generated by pumping ambient air through a 0.45-µm Teflon filter and fed the sampling train (i.e., $Na_2CO_3$ denuder and the wetted frit disc sampler) to generate the measurement baseline for $pNO_3$ measurement.

The manuscript is revised for a clearer description of the measurement systems.

Page 5-6: The separation of the terms pNO3 and NITs for particle nitrate measurements is confusing when later used for parameterization of the HONO source (pNO3). I suggest to use for both measurements pNO3 (particle nitrate…) and index the different instruments, e.g. pNO3(LPAP) and pNO3(filter).

We agree that this is somewhat confusing. In the revised manuscript, $pNO_3^{LPAP}$ represents particulate nitrate concentrations measured by the LPAP system and $pNO_3^{filter}$ represents particulate nitrate concentrations determined in the aerosol filter samples.

Line 149: The explanation of the correction factor of 2.06 (by the way: *2.06 or /2.06?) applied for the LPAP pNO3 measurements is not reasonable. An uptake of 50 % of the particles in the inlet of the frit sampler is too high (particles typically follow the gas stream, see principle of a denuder…). I expect more general differences between the two methods. For example, is $HNO_3$ (sticky…) also collected on the filter? If the $HNO_3$/pNO3 ratio is 1:1 the factor of two could be explained. In addition, was the pNO3 recovery efficiency of the frit sampler measured in laboratory studies? May be the nitrate collected on the frit is not completely extracted by water?

The correction should be applied as: $pNO_3^{LPAP}$ = 2.06*$pNO_3^{LPAP}$ (uncorrected). We edited the sentence in the revised manuscript to clarify.

We have to admit that the exact reason causing the discrepancy of the two $pNO_3$ measurements is not known. A collection efficiency determined for aerosol-phase ammonia in the frit-disk sampler was ≥ 99% (Huang et al., 2009), and it is reasonable to assume that aerosol-phase nitrate is quantitatively collected considering the high solubility of nitrate in water. The 50% difference could not be explained by the adsorption of $HNO_3$ on filters neither, since the $HNO_3$/$pNO_3$ ratio is quite low, with a median of 12% for our study. Loss of particles on the denuder is possible. The denuder that we used was originally designed for gas flow of 10 L/min, and the lower flow rate of 2 L min$^{-1}$ we used in the $pNO_3$-LPAP system might have resulted in some particle deposition.

Line 170: Was the fast reaction of NO+$O_3$ considered for in the 30 m long inlet line? At least for the plumes this is expected to change the NO/$NO_2$/$O_3$ system (quadratic reaction kinetics of NO+$O_3$).

The NO loss rate via NO+$O_3$ reaction is ~0.013 sec$^{-1}$, assuming a $k_{O3+NO}$ = 1.8*10$^{-14}$ cm$^3$ molecule$^{-1}$ s$^{-1}$ (Atkinson et al., 2004) and [$O_3$] of ~30 ppbv. With a tubing volume of 0.00024 m$^3$ (30 m of 1/8"-ID line) and a flow rate of 4 L min$^{-1}$, the residence time of air traveling in the sampling tubing is ~4 sec. Therefore, the decrease of NO concentration due to the NO+$O_3$ reaction in the 30-m long tubing is expected to be <5%.

Line 181: …Supplement Information S1.

Section number was added in the revised manuscript.

Line 182: the heading (pNO3) does not fit to the measurements (NITs), see above.

We agree that our notations for particulate nitrate were confusing. In the revised manuscript we use $pNO_3^{LPAP}$ for particulate nitrate concentrations measured by the LPAP system and $pNO_3^{filter}$ for particulate nitrate concentrations determined in the aerosol filter samples. We have reorganized the budget analyses in the revised manuscript and the original heading is no longer used.

Lines 196-197: Why was the HONO formation only measured for the short 5 min period? From my experience HONO formation decrease with time. Caused by the long lifetime of particles in

the atmosphere longer irradiation times (steady state…) are recommended in the laboratory. Thus, please mention that the EF(m) is an upper limit. Compare also former discussions on initial and steady state uptake coefficients.

The decrease HONO formation with time was not observed in this work under 5-min irradiation time. A decrease in HONO photoproduction under 10-min irradiation, which was observed by Ye et al. (2017), was not seen in this study neither. It is possible that HONO photoproduction might decrease under a longer irradiation time in laboratory settings. We disagree with the reviewer that long irradiation times should recommended for $pNO_3$ photolysis to meet the purpose of extrapolating lab-determined photolysis rate constants to ambient atmospheric environments. The decrease trend in HONO photoproduction likely result from changes in chemical properties of the aerosol samples while under irradiation in the photochemistry chamber. We suspect that the decreased HONO production might result from lose of protons or catalytic sites on the surface. We suggest measuring HONO formation during a short period so that the chemical properties of the aerosol sample remain relatively consistent with those freshly collected in the ambient environment.

Line 198: …Supplement Information S2…

Section number is added in the revised manuscript.

Table 1: If a correction factor was applied for pNO3 by comparison with NITs, why are the mean concentrations of pNO3 and NITs not exactly equal?

The correction factor was the slope of the linear correlation between the concentrations of $pNO_3$ and NITs (i.e., $pNO_3^{LPAP}$ and $pNO_3^{filter}$, respectively) that were measured simultaneously. However, there are nonoverlapping data points of the two measurements (Fig. 1 in the manuscript). The differences in the mean concentrations after the correction may mainly reflect the temporal variability in ambient $pNO_3$. Furthermore, the differences between $pNO_3^{filter}$ and $pNO_3^{LPAP}$ concentrations may also be in part attributed to the fact that the LPAP measurement system skipped ~17 min during each hour to sample the "zero-$pNO_3$" air, while each sample for $pNO_3^{filter}$ was continuously collected for four hours.

Figure 3: the relative patterns of HONO and $NO_x$ are very similar for spring and summer (island/plumes) in contrast to pNO3, which is another hint for $NO_2$ as main precursor of HONO.

Please see to our responses to major comments (1) and (2) above for our updated budget analyses of HONO daytime chemistry.

Lines 346-348: Please reformulate the sentence when P(extra) and not HONO is compared to the $NO_2$.

The comparison between $NO_2$ and HONO concentration that cannot be explained by NO+OH reaction is displayed in Fig. Re1 Further comparisons based on HONO formation rates are shown in Figs. Re2–Re4.   The results all support the conclusion that "$NO_2$ was unlikely a major HONO precursor in the background air masses when $NO_x$ was below 1 ppbv".

Lines 378-381: First, during night-time, aerosol reactions will not play any role (compare S/V of particles and the ground). Second, the HONO concentration alone (higher for island influenced compared to clean marine) is no argument for island surfaces being the main source region, since also the precursor ($NO_2$) is higher for the island influenced air masses. Here the HONO/$NO_2$ ratio should be considered! But also this ratio is higher for island influenced compared to clean marine conditions (HONO is factors of 4/7 higher, NO2 only a factor of 2-3) pointing to the heterogeneous HONO formation only on ground surfaces.

Agreed. We add texts in this paragraph in the revised manuscript to properly discuss and compare the diurnal patterns of HONO/$NO_2$ ratios in clean marine air and island-influenced air.

Equation (2): Instead of P($NO_x$->HONO), add P(NO+OH) and P($NO_2$+organics+hv).

As suggested, we add calculation in the revised manuscript for $P_{NO+OH \to HONO}$, $P_{NO_{2(aerosol)} \to HONO}$ for low-$NO_x$ environment (including clean marine and island-influenced air), and $P_{NO_{2(gound)} \to HONO}$ for island-influenced, low-$NO_x$ air masses (see our response to major comment (1) and (2) for details of the updated budget analyses).

Equation (4): Add the minor loss term L(HONO+OH).

As suggested, the calculations for $L_{HONO+OH}$ and $L_{deposition}$ are added in the revised manuscript.

Line 410: Please do not use the high $NO_x$ data to parameterize P($NO_2$+organics+hv) for low $NO_x$ conditions.

As suggested, we add calculation in the revised manuscript for $P_{NO+OH \to HONO}$, $P_{NO_{2(aerosol)} \to HONO}$ for low-$NO_x$ environment (including clean marine and island-influenced air), and $P_{NO_{2(gound)} \to HONO}$ for island-influenced, low-$NO_x$ air masses (see our response to major comment (1) and (2) for details of the updated budget analyses). We limit the discussion of HONO-$NO_2$ and HONO-$NO_x$ relationships only to the high-$NO_x$ plumes where $NO_x$ is the dominant HONO precursor.

Figure 5: Use only the clean marine and island influenced conditions to parameterize (P($NO_2$+organic+hv), compare to Fig. 4, where the variable plumes were also not considered. Besides the concentration dependence of the reaction, low HONO/$NO_x$ for high $NO_2$ could be also explained by different distances/transport times from the $NO_x$ sources to the measurement site, for which for low distance measured HONO/$NO_x$ will approach the HONO/$NO_x$ emission ratio, for which similar values of close below 1 % are known.

As suggested, we add calculation in the revised manuscript for $P_{NO+OH \to HONO}$, $P_{NO_{2(aerosol)} \to HONO}$ for low-$NO_x$ environment (including clean marine and island-influenced air), and $P_{NO_{2(gound)} \to HONO}$ for island-influenced, low-$NO_x$ air masses (see our response to major comment (1) and (2) for details of the updated budget analyses). We limit the discussion of Fig. 5 to the high-$NO_x$ plumes where $NO_x$ is the dominant HONO precursor.

Line 449: If HONO is deposited on the alkaline sea surfaces, then P(NO$_x$->HONO) should be the lower (…) limit. And do not use P(NO$_x$->HONO) in this way, see above.

Again as suggested, we calculate in the revised manuscript the production terms of $P_{NO+OH\rightarrow HONO}$, $P_{NO_{2(aerosol)}\rightarrow HONO}$ for clean marine air. Please see our response to major comment (1) for a detailed discussion.

Line 476-477: The most reasonable range of EF was 1-7 in Romer et al. In addition, the even lower values of Shi et al., 2021 are missing.

We edit the texts in the revised manuscript and make it clear to the reader that our $EF^m$ agree well with the low values reported by Romer et al. (2018) and Shi et al. (2021).

Line 482: Why were the data of NITs and not of pNO3 used here?

We choose to use the nitrate concentration determined from a half filter sample to calculate HONO production rate from particulate nitrate photolysis, since the photolysis rate constant was determined using the other half of the same filter sample. The nitrate concentration determined by the LPAP system is not used here. Some differences between pNO$_3$ $^{filter}$ and pNO$_3$ $^{LPAP}$ concentrations are expected, since the LPAP measurement system skipped ~17 min during each hour to sample the "zero-pNO$_3$" air, while each sample for pNO$_3$$^{filter}$ was continuously collected for four hours.

Equation (8): Use the experimental EF = 4, see laboratory studies.

Please see our response for major comment (4).

Line 519: Just a comment (out of this study…) to the high published EF values of Ye et al.: Could the increasing values with decreasing p(NO3) levels be an artificial background HONO formation of the reactor/filter set-up, which gets relatively more important for lower particle load?

For all of our irradiations, we made sure that HONO production from empty reaction chamber is negligible. We do observe HONO production from blank filters, and the produced HONO concentrations are usually quite low (<25 pptv) and have minimal effect on those high $J_{pNO_3}^N$ numbers reported by Ye et al. (2017). We also tested whether there exist artifacts from our experimental setup by irradiating samples collected from Delmar, New York, USA in fall 2020 and obtained $J_{pNO_3}^N$ values that ranged from 1.1×10$^{-4}$ to 6.6×10$^{-4}$ s$^{-1}$, which is in good agreement with those reported by Ye et al. (2017) for this exact sampling location ($J_{pNO_3}^N$ values ranged from 6.1×10$^{-5}$ to 4.3×10$^{-4}$ s$^{-1}$). This agreement gives us confidence that the large difference between $EF$ values reported by this work and Ye et al. (2017) does not result from experimental artifacts.

Lines 521-523: EF* is by definition (see above) systematically too large and EF(m) should be used (which may have some statistical uncertainties…).

Please see our response for major comment (4).

Figure 7: The high HONO and $NO_x$ levels observed before midnight 5/11 (dark no photolytic sink) may be simply explained by direct emissions, since the HONO/$NO_x$ ratio is only ca. 0.5 %, which is very close/even lower to published emission ratios. In contrast, the increasing HONO levels under similar $NO_x$ levels for the two marked plumes on the morning of 5/11 indicate again the photosensitized nature of the $NO_2$ conversion (increasing radiation).

We agree. These texts are added in the revised manuscript:

"Average HONO/$NO_x$ ratio was 0.6% and 0.7% in the high-$NO_x$ plumes before midnight and around sunrise of 5/11, respectively. These Low HONO/$NO_x$ ratios were close to the published emission HONO/$NO_x$ ratio of 0.79% (Liu et al., 2019), suggesting that direct anthropogenic emission was the dominant HONO source within the nighttime plume. It is interesting to point out that similar HONO/$NO_x$ ratios were also observed in the smaller plume observed around 9:00 am on 5/11, despite significant HONO photolytic loss during the air mass transport, indicating the photochemical enhancement in HONO formation from reactions (2) – (5)."

Lines 632-633: Modify that sentence when correctly evaluated, see above.

The conclusion section is revised according to our updated budget analyses.

Line 638: The EF of 30 is not "moderately enhanced" but systematically too high (see EF(m) = ca. 4).

An EF of 30 is "moderately enhanced" when compared to those high EF values reported by Ye et al. (2016b, 2017) and (Bao et al., 2018). The comparison between $P^*_{pNO_3 \rightarrow HONO}$ and $P^m_{pNO_3 \rightarrow HONO}$ in Fig. Re4 clearly illustrate that our EF* is significantly higher than our EF$^m$.

Line 640 – 642: Check after re-evaluation. I expect that the $NO_2$ reaction will get more important and the $HNO_3$ photolysis less important.

Please see our response for major comment (4).

Supplement:

Equation (S1): I do not understand the double normalization? The first normalization is clear, for which UVmodel is already the modelled clear sky UV radiation (= UV*model…). The modelled J-values are calculated for clear sky only, which may be in reality lower, caused by any clouds etc. To consider that, the ratio of measured and clear sky modelled UV is used. But why is the second normalization (UV*model/UV*measured) done?

The second normalization is to obtain a correction factor between the measured and modeled UV intensities at solar noon under clear sky, to correct systematic difference between the measured and modeled UV intensity. Actually, the second normalization does not affect our results

because the difference is smaller than 3%. The small difference is now added to the revised Supplement.

Line 35 and 36: are the units in the brackets "moles" or do you mean "molecules" (the latter is normally used in the paper)? For molecules use (molec.).

We mean "moles". The unit is consistent with those used in our previous publications for $pNO_3$ photolysis rate constants (Ye et al., 2017).

Line 39: $7x10^{-7}$ $s^{-1}$ should be for gaseous $HNO_3$ (and not nitrate).

Corrected in the revised Supplement.

**Technical corrections:**

We thank the reviewer for the time and effort spent in reviewing the technical details of our manuscript. These technical corrections are addressed in the revised manuscript.

Line 48: Li et al., 2008a (the first Li et al., 2008 in the text…), the second Li et al., 2008b is referred to in Line 74. Change accordingly in the reference list and the text.

Line 51: Dito for Ye et al., 2017a (is the first Ye et al. 2017 in the text…). In Line 327 it should be Ye et al., 2017b. Change accordingly in the reference list and the text.

Line 57: Acker et al., 2006a

Line 65: Villena et al., 2011a and in Line 165: Villena et al., 2011b. Change accordingly in the reference list and the text.

Line 81: Ye et al., 2017 a, b or both?

Line 123: … on the platform…

Line 130: Ye et al. (2016, 2018); dito in Line 456: Ye et al., 2016

Line 515: $NH_4NO_3$

References:

General:

- unify the doi: citation, either doi:… or https://doi.org.... (but not: doi:https://doi.org.);

- when the Science journal is cited always delete the bracket (80-. ).

Line 668: L02809 missing (see pdf of the paper)

Line 673: 336b

Line 689: 130,

Line 692: L02813 is missing

Line 699: D21202 is missing

Line 706: Becker, K. H.

Line 713: L05818 is missing

Line 722: L04803 is missing

Line 757: use 8192 and not the LOP numbers (see pdf of the paper)

Line 763: 4099 is missing

Line 766: 4705 is missing

Line 789: 1326-d

Line 809: L15820 is missing

Line 814: 4590 is missing (delete the ACH numbers, see pdf of the paper)

Line 817: 2217 is missing

Line 820: D08311 is missing

Equation S3: Should be [Salicylic acid]

**Reviewer #2**

General comments:

The paper titled "An investigation into the chemistry of HONO in the marine boundary layer at Tudor Hill Marine Atmospheric Observatory in Bermuda" by Zhu et al. present measurement results of temporal distributions of nitrous acid (HONO) and its budget analysis in background marine environments at Bermuda. Laboratory study examining the importance of particle nitrate as photolytic HONO source was also conducted along with the field campaigns. The photolytic of Pno3 and HNO3 were found to dominate HONO production in Bermuda, largely different from those reported in urban and polluted areas, which may suggest the unique chemistry of HONO as well as other reactive nitrogen species in the observation site, although these novel

results are still needed confirmed or further discussed. The manuscript was well written and presented, but some issues needed to be clarified. Therefore I recommend the publication of Zhu et al. work after replying the following comments clearly.

We sincerely thank the reviewer for the comments and suggestions, which help improving our manuscript. Our response to each reviewer query is colored in blue.

Specific and technical comments:

1. Line 25-26, As discussed in section 3.4, NOx-related reactions contributed minorly in daytime formation of HONO, while the photolytic of Pno3 and HNO3 dominate HONO production in marine and island influenced air, respectively. Please confirm it and revised it properly.

In Line 25–26 of the original manuscript, we stated that $NO_x$-related reactions played dominant roles in daytime formation of HONO in polluted plumes emitted from local traffic, power plant and cruise ship emissions, which was confirmed in section 3.4.1 in the original manuscript. Therefore, this sentence is not revised.

2. Table 1. The statistical result of pNO3 and NITs are better present in ug/m3, which was in particle phase. In addition, as bld values were obtained for HONO and NOx during the campaigns, the calculation for the mean values should be provided.

In this manuscript, the concentration of particulate nitrate is presented in pptv for direct comparison to the concentrations of other reactive nitrogen species (e.g., $NO_2$), and for conveniently calculating HONO production rate through particulate nitrate photolysis which is presented in pptv·s$^{-1}$. Detailed discussion regarding water-soluble ions in marine aerosol samples will be included in a separate manuscript (Zhu et al., in preparation), and we will take the reviewer's suggestion and present the concentrations of those water-soluble ions in µg/m$^3$.

The mean values for HONO, $NO_x$, $pNO_3$, NITs and HONO/$NO_x$ ratio were provided in Table 1 of the original manuscript.

3. Figure 2, it is interesting to note that the highest ratio of HONO/NOx appeared in the southeast direction, while both HONO and NOx showed highest values in the northeast direction, where the city of Hamilton located. Can you explain it?

We agree with the reviewer that this is an interesting finding. Actually, we divided our measurement datasets into three different categories (i.e., clean marine air, island-influenced air, and polluted plumes) based on the wind direction and the $NO_x$ level. The dependences of HONO, $NO_x$, and HONO/$NO_x$ ratios on wind direction are shown in Fig. 2 (please see section 3.1).

$NO_x$ was the dominant HONO precursor in the high-$NO_x$ plumes, which were mostly dominated by the northeasterly (i.e., from highly populated centers, power plant in Hamilton and cruise ships in Royal Naval Dockyard) and only accounted for small fraction of air masses modified by the island. On the other hand, the majority of the island-influenced air masses were from less populated area in the southeastern quarter and contained relatively low levels of $NO_x$ ($\leqslant$ 1 ppbv). Under the low $NO_x$ conditions, the photolysis of nitric acid/nitrate adsorbed ($HNO_{3(ads)}$) on island surfaces may become an important HONO source in addition to the $NO_x$-related reactions, resulting in higher HONO/$NO_x$ ratios than those observed in high-$NO_x$ polluted plumes. We revise the discussions in section 3.4.3 to remind the readers that the highest HONO/$NO_x$ ratios occurred in the low-$NO_x$ island-influenced air masses.

4. Line 255-260. The spikes that associated with the ship emissions should be carefully checked.

We agree with the reviewer that our description for the spikes that were removed in the clean marine air was not sufficient. The removed spikes represent average $NO_2$ and HONO concentrations of 1563 pptv and 15 pptv, respectively, and were significantly higher than the 75[th] percentile for $NO_2$ and HONO concentration in clean marine air (see Fig.3 in the original manuscript). We added the following sentence to the revised manuscript:

"Average concentrations for $NO_2$ and HONO within the removed spikes were 1563 pptv and 15 pptv, respectively."

5. Line 260-265 I do not think the contribution from direct emissions would be higher during night than during the day, as human activities were much reduced during the night, which supported by the diurnal pattern of NO2 in Fig.4. Higher NO2 concentrations were observed during the day. I agree that the transport times (1.4h in 2m/s wind speed) may longer than the photolytic lifetime of HONO in the daytime. However, one should note that the higher NO2 appeared in the early morning that the radiation in relative low levels, thus the photolytic lifetime of HONO expected to be much longer. Please clarify it.

The reviewer is correct that the direct emissions (based on the absolute emission rates) should be higher during the day than at the night. However, relative to the contributions from other processes, direct emissions were less important during the day than the night, since HONO productions through gaseous reaction between NO and OH, photosensitized heterogeneous reaction of $NO_2$ and the photolysis of nitric acid/nitrate on surfaces are present during the day and absent at night.

We also agree with the reviewer that direct emission might be important in the early morning plumes. The sentence is revised as follows:

"The relative contribution from direct emissions to the high-level HONO may be substantial during the night or under low-light conditions, but is expected to be relatively small within 4 hours before and after solar noon. Estimated transport times from the city of Hamilton to the THMAO site were $\geq$ 1.4 h (with ~10 km distance and ~2 m/s wind speed) and were several

times longer than the HONO photolytic lifetime (~10–18 min from 9:30 to 17:30 under clear skies) during the day."

6. Line 440-445. The using of equations (5) and (5') would introduce large uncertainty in estimating the P NOx-HONO, as the heterogeneous production from nitrogen dioxide (NO2) on serval surface is non-linear. Thus, the estimated missing source for HONO by equation (6) is questionable. In addition, direct emission was not included in equation (6) which may overestimate the Pmissing as mentioned above. In addition, I would expect the authors considered and evaluated the contributions of photosensitized heterogeneous conversion on island surfaces and aerosol surface that reported to be contributed importantly to HONO formation (i.e., Liu et al., 2021).

We agree with the reviewer that simplified estimates for HONO production rates from $NO_x$-related reactions in the original manuscript have significant limitations due to the non-linear relationship between the photosensitized reaction rate and $NO_2$ concentration, as well as changes in air mass types and conditions (e.g., with or without contact with ground surfaces). After considering both reviewers' suggestions, we estimate daytime HONO production rates from gas-phase NO+OH reaction and $NO_2$ heterogeneous reaction separately in the revised manuscript.

For clean marine air, equations Re2 and Re3 are used to estimate the HONO production rates through R2 and R3 in the original manuscript, and $k_{NO_2-aerosol}$ was calculated using equation Re4 (Please see our response to reviewer#1 for details). $P_{NOx \to HONO}$, defined here as the sum of $P_{NO+OH \to HONO}$ and $P_{NO_{2(aerosol)} \to HONO}$ is estimated as $9.0 \times 10^{-4}$ pptv·s$^{-1}$ in clean marine air, and is lower than our previous estimate of $1.4 \times 10^{-3}$ pptv·s$^{-1}$ by equation (5') in the original manuscript. Therefore, we concluded that our simplified estimates based on HONO-$NO_x$ relationship in polluted plumes overestimated HONO production rates in clean marine air, since the ocean surface does not support the formation of HONO via heterogeneous reaction from $NO_2$.

For island-influenced air, we estimate the rate and rate constant for $NO_2$ to HONO conversions on ground surfaces ($P_{NO_{2(gound)} \to HONO}$) with equations Re6 and Re7 (Please see our response to reviewer#1 for details). The calculated median of $P_{NO_2-ground}$ in island-influenced air is $7.0 \times 10^{-3}$ pptv·s$^{-1}$ and accounts for 28% of the HONO daytime production budget in island-influenced air masses.

We agree with the reviewer that our original estimates for $P_{missing}$ is problematic in the original manuscript due to large uncertainties associated with our original estimates for $P_{NOx \to HONO}$. In the revised manuscript, our calculated medians for $P_{NO+OH \to HONO}$ and $P_{NO_{2(aerosol)} \to HONO}$ are $8.9 \times 10^{-4}$ and $6.0 \times 10^{-6}$ pptv·s$^{-1}$ in clean marine air, which only accounted for minor fractions (21% and 0.14%, respectively) of the median HONO production rate needed ($4.2 \times 10^{-3}$ pptv·s$^{-1}$) to sustain HONO loss rate, indicating that a large fraction (~79%) of HONO production rate is missing after counting the contributions from $NO_x$-related reactions. In island-influenced air, 62% of HONO production rate ($1.6 \times 10^{-2}$ pptv·s$^{-1}$) is still missing after counting the contributions from $P_{NO+OH \to HONO}$ ($2.6 \times 10^{-3}$ pptv·s$^{-1}$), $P_{NO_{2(aerosol)} \to HONO}$ ($2.0 \times 10^{-5}$ pptv·s$^{-1}$), and

$P_{NO_2(gound)\rightarrow HONO}$ ($7.0\times10^{-3}$ pptv·s$^{-1}$), indicating that NO$_x$-related reactions are not sufficient to support the HONO concentrations.

Direct emission is not included in our budget analyses for HONO daytime chemistry, since we only analyzed data that were measured from 10:00 to 15:00. During this time period, the average photolytic lifetime of HONO is 11 min, which is lower than the air transport time from the city of Hamilton to the sampling site (~1.4 h) by a factor of 7.6. Therefore, the observed HONO in the daytime plumes at the site was mostly produced during the transport of the air masses from emission sources to the site via gas-phase reaction, heterogeneous reactions on aerosol and island surfaces, and maybe some other chemical sources.

7. Line 515. HN4NO3 or NH4NO3? In addition, as the water-soluble ions were also analyzed, the existence of NO3- in particles should be discussed, which would further support the comparable of low EF values from this study with that reported by Shi et al., 2021.

Should be NH$_4$NO$_3$. We thank the reviewer for pointing out this typing error. We agree with the reviewer that discussing the nitrate loading on particles is important and that it is necessary to make direct comparisons to other research works that report laboratory-determined particulate nitrate photolysis rate constants. Additional details for the temporal variations and potential factors (including nitrate loading) affecting pNO$_3$ photolysis rate constants are to be discussed in a separate manuscript (Zhu et al., in preparation).

8. Line 521-523, I note that an upper limit EF* of 30 was used, not the measured EF(m) of much lower value (about 4) determined in the laboratory of this study, if it is true claimed by the author that using the store aerosol samples caused large discrepancy, how about the results of Pno3 photolysis rate conducted by previous laboratory studies using similar method? I do not think the using of high EF*that result in a high contribution of Pno3 photolysis on HONO production is reasonable. At least, the authors should evaluate the results using the high and low EF values.

We agree with the reviewer that we should present both *EF\** calculated from field observation data and *EF$^m$* obtained directly from laboratory measurements and compare HONO production rate calculated using *EF\** and *EF$^m$*. In the revised manuscript, we modify the discussion and add a figure (Fig. Re4, presented in our response to reviewer #1) to compare our estimated HONO production rates by equations Re9 and Re12 (see our response to reviewer #1).

The added Fig. Re4 clearly shows the large difference in our estimates for HONO production rates via pNO$_3$ photolysis using *EF\** and *EF$^m$*. We also revised the text to point out to the reader that *EF\** is an upper-limit enhancement factor for pNO$_3$ photolysis relative to HNO$_3$ photolysis.

To explain the large difference between *EF\** and *EF$^m$*, we pointed out in the original manuscript potential biases in lab-determined $J^N_{pNO_3}$, caused by changes in aerosol properties during aerosol sample collection and storage, such as aggregation of particles and deprotonation of nitrate. It should be pointed out that pNO$_3$ concentration and

pNO$_3$/HNO$_3$ ratio (an indicator for aerosol acidity) in the lower marine boundary layer in this work were significantly higher than in the upper marine boundary layer (Ye et al., 2016b) and in the terrestrial boundary layer (Ye et al., 2017), resulting in the determined lower $J^N_{pNO_3}$ values.

9. Line 545-549. HONO production rate of 0.016 pptv/s from photolytic of HNO3 was suggested, which contributed 84% of the HONO photolytic loss rate. How this value is calculated, and is it possible for such high contribution of HNO3 photolytic on HONO which rarely reported? In addition, how about its contribution in marine influenced air? I note that this value was not provide in Table 2.

In the original manuscript, we provided an upper-limit estimate for HONO production through the photolysis nitric acid on island surfaces ($P_{HNO_{3(ads)}\rightarrow HONO}$) by assuming it accounted for 100% of the HONO production rate unaccounted for by NO$_x$-related reactions and pNO$_3$ photolysis ($P_{unaccounted}$). We agree with the reviewer (and reviewer #1) that $P_{HNO_{3(ads)}\rightarrow HONO}$ is unlikely responsible for 100% of $P_{unaccounted}$, since there is a high intercept (~5.6 pptv) in the [HONO]$_{unaccounted}$ versus [HNO$_3$]$_{ave}$ plot (Fig. Re5 in our response to reviewer #1). Therefore, in the revised manuscript, we report $P_{unaccounted}$ as the sum of $P_{HNO_{3(ads)}\rightarrow HONO}$ and $P_{other}$, where is the HONO production rate by other processes that are not included in the budget analysis

An estimate for $P_{HNO_{3(ads)}\rightarrow HONO}$ is not performed for the clean marine air masses coming directly from open ocean; the surface of alkaline seawater serves as a sink for HONO and HNO$_3$ and does not support the production of HONO through photolysis of $HNO_{3(ads)}$.

References:

Atkinson, R., Baulch, D. L., Cox, R. A., Crowley, J. N., Hampson, R. F., Hynes, R. G., Jenkin, M. E., Rossi, M. J. and Troe, J.: Evaluated kinetic and photochemical data for atmospheric chemistry: Volume I - gas phase reactions of O$_x$, HO$_x$, NO$_x$ and SO$_x$ species, Atmos. Chem. Phys., 4(6), 1461–1738, doi:10.5194/acp-4-1461-2004, 2004.

Bao, F., Li, M., Zhang, Y., Chen, C. and Zhao, J.: Photochemical aging of beijing urban PM2.5: HONO production, Environ. Sci. Technol., 52(11), 6309–6316, doi:10.1021/acs.est.8b00538, 2018.

Crilley, L. R., Kramer, L. J., Pope, F. D., Reed, C., Lee, J. D., Carpenter, L. J., Hollis, L. D. J., Ball, S. M. and Bloss, W. J.: Is the ocean surface a source of nitrous acid (HONO) in the marine boundary layer?, Atmos. Chem. Phys., 21(24), 18213–18225, doi:10.5194/acp-21-18213-2021, 2021.

Harrison, R. M., Peak, J. D. and Collins, G. M.: Tropospheric cycle of nitrous acid, J. Geophys. Res. Atmos., 101(D9), 14429–14439, doi:10.1029/96JD00341, 1996.

Huang, G., Hou, J. and Zhou, X.: A Measurement method for atmospheric ammonia and primary amines based on aqueous sampling, OPA derivatization and HPLC analysis, Environ. Sci.

Technol., 43(15), 5851–5856, doi:10.1021/es900988q, 2009.

Li, G., Lei, W., Zavala, M., Volkamer, R., Dusanter, S., Stevens, P. and Molina, L. T.: Impacts of HONO sources on the photochemistry in Mexico City during the MCMA-2006/MILAGO Campaign, Atmos. Chem. Phys., 10(14), 6551–6567, doi:10.5194/acp-10-6551-2010, 2010.

Liu, Y., Nie, W., Xu, Z., Wang, T., Wang, R., Li, Y., Wang, L., Chi, X. and Ding, A.: Semi-quantitative understanding of source contribution to nitrous acid (HONO) based on 1~year of continuous observation at the SORPES station in eastern China, Atmos. Chem. Phys., 19(20), 13289–13308, doi:10.5194/acp-19-13289-2019, 2019.

Romer, P. S., Wooldridge, P. J., Crounse, J. D., Kim, M. J., Wennberg, P. O., Dibb, J. E., Scheuer, E., Blake, D. R., Meinardi, S., Brosius, A. L., Thames, A. B., Miller, D. O., Brune, W. H., Hall, S. R., Ryerson, T. B. and Cohen, R. C.: Constraints on aerosol nitrate photolysis as a potential source of HONO and $NO_x$, Environ. Sci. Technol., 52(23), 13738–13746, doi:10.1021/acs.est.8b03861, 2018.

Shi, Q., Tao, Y., Krechmer, J. E., Heald, C. L., Murphy, J. G., Kroll, J. H. and Ye, Q.: Laboratory investigation of renoxification from the photolysis of inorganic particulate nitrate, Environ. Sci. Technol., 55(2), 854–861, doi:10.1021/acs.est.0c06049, 2021.

Stemmler, K., Ammann, M., Donders, C., Kleffmann, J. and George, C.: Photosensitized reduction of nitrogen dioxide on humic acid as a source of nitrous acid, Nature, 440(7081), 195–198, doi:10.1038/nature04603, 2006.

Stutz, J., Alicke, B. and Neftel, A.: Nitrous acid formation in the urban atmosphere: Gradient measurements of NO2 and HONO over grass in Milan, Italy, J. Geophys. Res. Atmos., 107(D22), 8192, doi:10.1029/2001JD000390, 2002.

Ye, C., Gao, H., Zhang, N. and Zhou, X.: Photolysis of nitric acid and nitrate on natural and artificial surfaces, Environ. Sci. Technol., 50(7), 3530–3536, doi:10.1021/acs.est.5b05032, 2016a.

Ye, C., Zhou, X., Pu, D., Stutz, J., Festa, J., Spolaor, M., Tsai, C., Cantrell, C., Mauldin, R. L., Campos, T., Weinheimer, A., Hornbrook, R. S., Apel, E. C., Guenther, A., Kaser, L., Yuan, B., Karl, T., Haggerty, J., Hall, S., Ullmann, K., Smith, J. N., Ortega, J. and Knote, C.: Rapid cycling of reactive nitrogen in the marine boundary layer, Nature, 532(7600), 489–491, doi:10.1038/nature17195, 2016b.

Ye, C., Zhang, N., Gao, H. and Zhou, X.: Photolysis of particulate nitrate as a source of HONO and $NO_x$, Environ. Sci. Technol., 51(12), 6849–6856, doi:10.1021/acs.est.7b00387, 2017.

Zhang, N., Zhou, X., Shepson, P. B., Gao, H., Alaghmand, M. and Stirm, B.: Aircraft measurement of HONO vertical profiles over a forested region, Geophys. Res. Lett., 36(15), L15820, doi:10.1029/2009GL038999, 2009.

Zhou, X., Civerolo, K., Dai, H., Huang, G., Schwab, J. and Demerjian, K.: Summertime nitrous acid chemistry in the atmospheric boundary layer at a rural site in New York State, J. Geophys. Res. Atmos., 107(D21), 4590, doi:10.1029/2001JD001539, 2002.

---

## Author Response (AR1)

**Response to Reviewers**

**An investigation into the chemistry of HONO in the marine boundary layer at Tudor Hill Marine Atmospheric Observatory in Bermuda**

by Yuting Zhu and Xianliang Zhou

**Reviewer #1**

In the manuscript by Zhu et al., the chemistry of HONO at a marine measurement site on Bermuda island was investigated. Different HONO sources were discussed for which NOx related reactions were considered important under polluted island influenced conditions, whereas HONO formation by particle nitrate photolysis was found to be more important under clean marine conditions. Furthermore, photolysis of ground surface adsorbed HNO3 was postulated as main HONO source reaction for low NOx island influenced conditions. An important observation is the missing night-time formation of HONO under clean marine conditions, which is reasonable considering the alkaline sea surfaces acting as a perfect sink for HONO. This result agrees with another recent paper (Crilley et al., ACP, 2021, not cited), but is in contrast to former observations from China and Canada.

I have several comments, which should be considered in the revised paper.

We sincerely thank the reviewer for the detailed comments and the helpful suggestions, which help improving our manuscript. Our responses to the reviewer's comments are colored in blue below.

We appreciate that the reviewer pointed out that Crilley et al. (2021) was not cited in the original manuscript. This recent paper is now cited in the revised manuscript.

**Major Comments:**

1) Description of the HONO daytime budget:

For the production rate (equation 2) NOx related HONO formation is not well described. First, HONO formation by the gas phase reaction NO+OH could be easily implemented by assuming a reasonable diurnal OH profile (e.g. by its correlation with  $J(O^1D)$ , or by any simple box model). Since the homogeneous formation is most probably not too important here, even large uncertainties (e.g. factor of two...) will not matter too much. When OH is calculated, also the (minor) loss of HONO by its reaction with OH could be explicitly considered besides the HONO photolysis (see equation 4, where this reaction is now neglected). Considering OH, the PSS of HONO can be calculated and only excess levels (and not measured HONO...) should be explained by the discussed sources (Pextra).

Second and much more important, HONO formation by NO2 conversion during daytime must be implemented as a photolytic process, which is typically parameterized as a function of the NO2 concentration and (!) J(NO2), see work by Stemmler et al.. In this context, also reaction (R3) must include its photosensitized character! Please add hv to the reaction (compare R4). A dark reaction R3 is also known, but since night-time HONO formation was not observed, this is not of importance here. When the photosensitized conversion is correctly included, most probably the diurnal shape of the HONO production (Pextra) will be well explained (see below). Thus, the different patterns of HONO (one daytime maximum) and NO2 (two maxima) (see lines 346-348 and Fig. 4) cannot demonstrate the missing importance of reaction (R3) as long as this reaction is not correctly considered.

We observed a strong correlation between the noontime concentrations of HONO and  $NO_x$  under high-NOx conditions in this study. The observed HONO-NOx relationships were extrapolated to low-NOx conditions in the original manuscript for a simplified estimates for HONO production rates from NOx-related reactions. We agree with the reviewer that this simplified approach has significant limitations due to the non-linear relationship between the photosensitized reaction rate and NO2 concentration, as well as changes in air mass types and conditions (e.g., with or without contact with ground surfaces).

We follow the reviewer's suggestions and estimate daytime HONO production rates from gasphase NO+OH reaction and NO2 heterogeneous reaction separately in the revised manuscript, in the following HONO budget equation:

$$\frac{d[HONO]}{dt} \approx 0$$

$$= P_{NO+OH \to HONO} + P_{NO_{2}(aerosol)} \to HONO} + P_{NO_{2}(ground)} \to HONO} + P_{pNO3 \to HONO} + P_{HNO3(ads)} \to HONO} + P_{other} - (L_{photolysis} + L_{HONO+OH} + L_{deposition})$$
(Re1)

For clean marine air, the following equations are used to estimate the HONO production rates through R2 and R3 in the original manuscript:

$$P_{NO+OH \to HONO} = k_{NO+OH} \times [OH] \times [NO]$$
(Re2)

$$P_{NO_{2}(aerosol)} \rightarrow HONO} = k_{NO_{2}-aerosol} \times [NO_{2}]$$
(Re3)

Where  $k_{NO+OH}$  is the reaction rate constant between NO and OH obtained from Atkinson et al. (2004), and a constant [OH] of  $6 \times 10^6$  molecules cm-3 is assumed. For equation Re3,  $k_{NO_2-aerosol}$  is calculated using the following equation:

$$k_{NO_2-aerosol} = \frac{1}{4} \times \overline{v_{NO_2}} \times \frac{s}{v} \times \gamma_{NO_2-aerosol}$$
(Re4)

Where  $\overline{\upsilon_{NO_2}}$  is the average molecular speed of NO2,  $\frac{s}{v}$  is the surface area to volume ratio, and  $\gamma_{aerosol}$  is the uptake coefficient of NO2 on aerosol surfaces. A  $\frac{s}{v}$  ratio of 5×10-5 m-1 is used, based on 20 µg·m-3 of 1-µm sea-salt aerosol particles. An upper limit  $\gamma_{NO_2-aerosol}$  value of 2×10-5 is also used, taking account the photo-enhancement of HONO formation through heterogeneous reaction of NO2 (Li et al., 2010; Stemmler et al., 2006). The calculated

medians for  $P_{NO+OH\rightarrow HONO}$  and  $P_{NO_{2}(aerosol)\rightarrow HONO}$  are  $8.9 \times 10^{-4}$  and  $5.9 \times 10^{-6}$  pptv·s-1 in clean marine air, which only account for minor fractions (21% and 0.14%, respectively) of the median HONO production rate needed to sustain HONO photolytic loss. A large fraction (~79%) of HONO production rate cannot be accounted for by the contribution from NOxrelated reactions.  $P_{NOx\rightarrow HONO}$ , defined here as the sum of  $P_{NO+OH\rightarrow HONO}$  and  $P_{NO_{2}(aerosol)\rightarrow HONO}$ is estimated to be  $9.0 \times 10^{-4}$  pptv·s-1 in clean marine air, is lower than our previous estimate of  $1.4 \times 10^{-3}$  pptv·s-1 by equation (5') in the original manuscript.

We also agree with the reviewer that the daytime production of HONO from NO2 heterogeneous reaction is more important as a HONO source in island-influenced air than clean marine air. We calculate the HONO concentration that cannot be explained by gaseous production ([HONO]unexplained]):

$$[HONO]_{unexplained} = [HONO] - \frac{P_{NO+OH \to HONO}}{J_{HONO} + k_{HONO+OH} \times [OH] + \frac{\nu_{HONO}}{H}}$$
(Re5)

Where  $J_{HONO}$  is the HONO photolysis rate constant,  $k_{HONO+OH}$  is the HONO+OH reaction rate constant,  $v_{HONO}$  is the deposition velocity, which is set to a high value of 1 cm s-1 (Harrison et al., 1996; Stutz et al., 2002), and *H* the vertical transport distance. The value of *H* (i.e., 116 m) was calculated following Zhang et al. (2009), with an assumed turbulent diffusion coefficient Kz of 105 cm2 s-1 and a HONO photolytic lifetime of 670 s (11 min). [HONO]unexplained is then plotted against [NO2] in island-influenced air (Panel A) and clean marine air (Panel B) in Fig. Re1. A slight correlation was found for island-influenced air (R2=0.23), and no correlation was observed for clean marine air (R2=0.03), supporting the arguments that heterogeneous NO2 reaction contribute to a certain fraction of observed HONO concentration in the island-influenced air, but not in the clean marine air.

**Figure Re1:** [HONO]unexplained plotted against [NO2] in (A) island influenced air and (B) clean marine air from 10:00 to 15:00. The dash lines indicate the best-fit lines for linear regression between the two parameters in each figure.

We use these following equations to estimate the rate and rate constant for NO2 to HONO conversions on ground surfaces ( $P_{NO_{2(gound)} \rightarrow HONO}$ ):

$$P_{NO_{2}(gound)} \rightarrow HONO} = k_{NO_{2}-ground} \times [NO_{2}]$$
(Re6)

$$k_{NO_2-ground} = \frac{1}{4} \times \overline{v_{NO_2}} \times \frac{s}{v} \times \gamma_{NO_2-ground}$$
(Re7)

An S/V value of 0.017 m-1 was calculated with an effective surface area of 2 m2 per geometric surface area and an air column height (*H*) of 116 m. The value of *H* was calculated following Zhang et al. (2009) assuming a turbulent diffusion coefficient Kz of 105 cm2 s-1, and a HONO photolytic lifetime of 670 s (11 min).  $\gamma_{NO_2-ground}$  is set as  $2 \times 10^{-5}$ , which is a upper-limit value suggested by Stemmler et al. (2006). The calculated median of  $P_{NO_2-ground}$  in island-influenced air is  $7.0 \times 10^{-3}$  pptv·s-1 and accounts for 28% of the HONO daytime production budget in island-influenced air masses.

We would like to thank the reviewer for pointing out that  $\gamma_{NO_2}$  is expected to decrease with NO2 concentration. For this work, NO2 heterogeneous reactions are only parameterized for low-NOx conditions ([NO2] < 1 ppbv). We choose to use the upper-limit  $\gamma_{NO_2}$  value of  $2 \times 10^{-5}$  suggested by Stemmler et al. (2006) because our NO2 level is at the lower end of the NO2 concentration range in Stemmler et al. (2006), and significant decrease in  $\gamma_{NO_2}$  is not expected when NO2 concentration is below 1 ppbv.

The calculation results above indicate that heterogeneous production of HONO from NO2 is more important as a HONO source in island-influenced air than clean marine air, as suggested by the reviewer. However, it should be noted that the combined HONO production from NOxrelated reactions is only a minor HONO source even in island-influenced air; 62% of HONO production rate is still missing after counting the contributions from PNO+OH→HONO, PNO2(aerosol)→HONO, and PNO2(gound)→HONO.

**2) Proposed extra sources:**

To identify potential HONO formation mechanisms during daytime, only the extra HONO sources (besides the known reaction NO+OH, and the know losses by photolysis and reaction with OH, see above) should be calculated and the diurnal profile of  $P_{extra}$  be compared with different postulated sources, e.g. by plotting against different parameters, like J(NO2)xNO2, J(HNO3)xHNO3(av), J(HNO3)xpNO3 etc. for clean marine and island influenced conditions. I am quite sure that the best correlation will be obtained for J(NO2)xNO2 for island influenced conditions (see below). This will be in contrast to the statement made simply from the concentration (...) profiles (see lines 346-348), which does not include the photolytic nature of the formation process by NO2 reactions. In addition, there is always a non-linear connection between the HONO production and its concentration, caused by the variable main loss term (photolysis). In addition, reaction R3 is underestimated and R4 overestimated, see below.

HONO daytime production rate that cannot be explained by the gaseous reaction between NO and OH ( $P_{unexplained}$ ) can be calculated as:

$$P_{unexplained} = [HONO] \times (J_{HONO} + k_{HONO+OH} \times [OH] + \frac{v_{HONO}}{H}) - P_{NO+OH \to HONO}$$
(Re8)

Here we follow the reviewer's suggestion and plot  $[NO_2] \times J_{NO2}$ ,  $[pNO_3] \times EF^* \times J_{HNO3}$ , and  $[HNO_3]_{ave} \times J_{HNO3}$  versus  $P_{unexplained}$  in island-influenced air (Fig. Re2).  $[HNO_3]_{ave} \times J_{HNO3}$  exhibit better correlation (R2 =0.34) with Punexplained than  $[NO_2] \times J_{NO2}$  (R2 = 0.27) and much better than  $[pNO_3] \times EF^* \times J_{HNO3}$  (R2 = 0.16).